# EControl: Fast Distributed Optimization with Compression and Error Control

**Yuan Gao**[1,3]* **Rustem Islamov**[2]* **Sebastian U. Stich**[3]
[1]Universität des Saarlandes, [2]Universität Basel, [3]CISPA Helmholtz Center for Information Security
`{yuan.gao, stich}@cispa.de, rustem.islamov@unibas.ch`

## Abstract

Modern distributed training relies heavily on communication compression to reduce the communication overhead. In this work, we study algorithms employing a popular class of contractive compressors in order to reduce communication overhead. However, the naive implementation often leads to unstable convergence or even exponential divergence due to the compression bias. Error Compensation (EC) is an extremely popular mechanism to mitigate the aforementioned issues during the training of models enhanced by contractive compression operators. Compared to the effectiveness of EC in the data homogeneous regime, the understanding of the practicality and theoretical foundations of EC in the data heterogeneous regime is limited. Existing convergence analyses typically rely on strong assumptions such as bounded gradients, bounded data heterogeneity, or large batch accesses, which are often infeasible in modern Machine Learning Applications. We resolve the majority of current issues by proposing EControl, a novel mechanism that can regulate error compensation by controlling the strength of the feedback signal. We prove fast convergence for EControl in standard strongly convex, general convex, and nonconvex settings without any additional assumptions on the problem or data heterogeneity. We conduct extensive numerical evaluations to illustrate the efficacy of our method and support our theoretical findings.

## 1 Introduction

The size of modern neural networks has increased dramatically. Consequently, the data required for efficient training is huge as well, and accumulating all available data into a single machine is often infeasible. Because of these considerations, the training of large language models (Shoeybi et al., 2019), generative models (Ramesh et al., 2021; 2022), and others (Wang et al., 2020) is performed in a distributed fashion with decentralized data. Another quickly developing instance of distributed training is Federated Learning (FL) (Konečný et al., 2016; Kairouz et al., 2019) where the goal is to train a single model directly on the devices keeping their local data private.

The communication bottleneck is a main factor that limits the scalability of distributed deep learning training (Seide et al., 2014; Strom, 2015). Methods that use lossy compression have been proposed as a remedy, with great success (Seide et al., 2014; Alistarh et al., 2017). It has been observed that *error compensation* (EC) mechanisms are crucial to obtain high compression ratios (Seide et al., 2014; Stich et al., 2018), and variants of these techniques (Vogels et al., 2019) are already integrated in standard deep learning frameworks such as PyTorch (Paszke et al., 2019) and been successfully used in the training of transformer models (Ramesh et al., 2021).

These methods were primarily developed for data center training scenarios where training data is shuffled between nodes. This data uniformity was also a limiting assumption in early analyses for distributed EC (Cordonnier, 2018; Stich & Karimireddy, 2020). To make compression suitable for training beyond data centers, such as federated learning, it is essential to take data heterogeneity (also termed *client drift* or *bias*) (Karimireddy et al., 2020; Mishchenko et al., 2019) into account. Mishchenko et al. (2019) designed a method for this scenario, but it can only support the restrictive class of unbiased compressors. It turned out, that handling arbitrary compressors is a challenge. A

---

*Equal contribution.

line of work developed the EF21 algorithm (Richtárik et al., 2021) that can successfully handle the bias/drift, but cannot obtain a linear speedup in parallel training, i.e. the training time does not decrease when more devices are used for training.

One of the main difficulties in developing a method that simultaneously combats client drift and maintains linear acceleration has been the complicated interaction between the mechanisms of bias and error correction. In this work, we propose EControl, a novel mechanism that can regulate error compensation by controlling the strength of the feedback signal. This allows us to overcome both challenges simultaneously. Our main contributions can be summarized as:

**New Method.** We propose a novel method EControl that provably converges (i) with arbitrary contractive compressors, (ii) under arbitrary data distribution (heterogeneity), (iii) and obtains linear parallel speedup.

**A Practical Algorithm.** EControl does not need to resort to impractical theoretical tricks, such as large batch sizes or repeated communication rounds, but instead is a lightweight extension that can be easily added to existing EC implementations.

**Fast Convergence.** We demonstrate the convergence guarantees in all standard regimes: strongly convex, general convex, and nonconvex functions. In all the cases complexities are asymptotically tight with stochastic gradients. In the nonconvex case, the rate in the noiseless setting matches the known lower bound (He et al., 2023). In the strongly convex and general convex settings, we achieve the standard linear and sublinear convergence respectively, with tight dependency on the compression parameter in the noiseless setting. To the best of our knowledge, our work is the first demonstrating that in strongly convex and general convex regimes without additional assumptions on the problem.

**Empirical Study.** We conduct extensive empirical evaluations that support our theoretical findings and show the efficacy of the new method.

## 2 COMMUNICATION BOTTLENECK AND ERROR COMPENSATION

In our work, we analyze algorithms combined with compressed communication. In particular, we consider methods utilizing practically helpful contractive compression operators.

**Definition 1.** *We say that a (possibly randomized) mapping* $\mathcal{C} \colon \mathbb{R}^d \to \mathbb{R}^d$ *is a contractive compression operator if for some constant* $0 < \delta \leq 1$ *it holds*

$$\mathbb{E}\left[\|\mathcal{C}(\mathbf{x}) - \mathbf{x}\|^2\right] \leq (1 - \delta) \|\mathbf{x}\|^2 \quad \forall \mathbf{x} \in \mathbb{R}^d. \tag{1}$$

The classic example satisfying (1) is Top-K (Stich et al., 2018), which preserves the K largest (in absolute value) entries of the input, and zeros out the remaining entries. The class of contractive compressors also includes sparsification (Alistarh et al., 2018; Stich et al., 2018) and quantization operators (Wen et al., 2017; Alistarh et al., 2017; Bernstein et al., 2018; Horváth et al., 2019), and many others (Zheng et al., 2019; Beznosikov et al., 2020; Vogels et al., 2019; Safaryan et al., 2022; Islamov et al., 2023). In this section, we review related works on error compensation, with a focus on the works that also consider contractive compressors.

### 2.1 SELECTED RELATED WORKS ON ERROR COMPENSATION

The EC mechanism Seide et al. (2014) was first analyzed for the single worker case in Stich et al. (2018); Karimireddy et al. (2019). Extensions to the distributed setting were first conducted under the assumption of homogeneous (IID) data, a constraint imposed either implicitly by assuming bounded gradients Cordonnier (2018); Alistarh et al. (2018) or explicitly without the bounded gradient assumption Stich & Karimireddy (2020). Choco-SGD was designed for communication compression over arbitrary networks and analyzed under similar IID assumptions Koloskova et al. (2019; 2020a). The analyses of distributed EC where further developed in Lian et al. (2017); Beznosikov et al. (2020); Stich (2020).

The DIANA algorithm by Mishchenko et al. (2019) was proposed as a solution for the heterogeneous data case, and introduced a mechanism that was able to account for the drift/bias on nodes with different data, but only when unbiased compressors were used. However, their result inspired many

follow-up works that combined contractive compressors with their bias correction mechanism (which requires an additional unbiased compressor that doubles the communication cost of the algorithm per iteration), such as Gorbunov et al. (2020); Stich (2020); Qian et al. (2021b).

Recently, Richtárik et al. (2021) introduced EF21 that fully supports contractive compressors and presented strong analysis in the full gradient (i.e. noiseless) regime. Nevertheless, the main problem of EF21 is weak convergence guarantees in the stochastic regime, i.e. when clients have access to stochastic gradient estimators only. The analysis of EF21 and its modifications require large batches and do not have linear speedup with $n$. Moreover, they demonstrate poor dependency on the compression parameter $\delta$. Zhao et al. (2022) improves the dependency on the compression parameter $\delta$ in a nonconvex setting but still requires large batches. Recently, Fatkhullin et al. (2023) resolved those issues by introducing a momentum-based variant of EF21. They demonstrate asymptotically optimal complexity for nonconvex losses.

A line of work studied accelerated methods with communication compression (Li et al., 2020; Li & Richtárik, 2021; Qian et al., 2021a; 2023; He et al., 2023), each with different additional requirements on the problem, the compressor class, or stochasticity and batch size of the gradient oracle. These methods show accelerated rates matching lower bounds in some regimes but are typically impractical due to those requirements and many parameter tuning.

## 2.2 Existing problems with Error Compensation

Below we list the main problems of existing theoretical results and highlight a historical overview of the main existing theoretical results in Table 1.

**Additional Communication.** Gorbunov et al. (2020); Stich (2020) modify the original EC mechanism following the DIANA method. However, their approach requires an additional unbiased compressor. Such an idea allows for building a better sequence of compressed gradient estimators but with a doubled per-iteration communication cost.

**Strong Assumptions.** Many earlier theoretical results for EC require strong assumptions, such as either the bounded gradient assumption (Koloskova et al., 2019; 2020a) or globally bounded dissimilarity assumption (Lian et al., 2017; Huang et al., 2022; He et al., 2023; Li & Li, 2023)[1]. Besides, Makarenko et al. (2023) analyses EF21-based algorithm under bounded iterates assumption.

**Large Batches.** Fatkhullin et al. (2023) gives an example where EF21 fails to converge with small batch size. NEOLITHIC (Huang et al., 2022; He et al., 2023) matches lower bounds with large batch requirements combined with multi-stage execution for hyperparameter tuning. These requirements make these methods less suitable for DL training, where small batches are known to improve generalization performance and convergence (LeCun et al., 2012; Wilson & Martinez, 2003; Keskar et al., 2017).

**Suboptimal Convergence Rates.** Current theoretical analysis of distributed algorithms combined with contractive compressors do not match known lower bounds in the nonconvex regime (Huang et al., 2022; He et al., 2023). Zhao et al. (2022); Fatkhullin et al. (2021) do not achieve a speedup in $n$ in the asymptotic regime while in contrast Koloskova et al. (2020a) has suboptimal rates in the low noise regime (e.g. full gradient computations). We are aware of only the works by Fatkhullin et al. (2023) and He et al. (2023), but the latter requires large batches at each iteration.

**Missing a Practical Method for the Convex Regime.** He et al. (2023) analyzed the accelerated NEOLITHIC in general convex and strongly convex regimes with large batch requirements. In fact, this method is mostly of a theoretical nature as the choice of the optimal parameters relies on the gradient variance at the solution, and it was shown to be slow in practice (Fatkhullin et al., 2023). To the best of our knowledge, there is no distributed algorithm utilizing only a contractive compression operator and provably converging in general convex and strongly convex regimes under standard assumptions.

In our work, we propose a novel method, which we call EControl, and push the theoretical analysis of algorithms combined with contractive compression operators further in several directions.

---

[1]This assumption heavily restricts the class of functions and typically does not hold in Federated Learning.

Table 1: Theoretical comparison of error compensated algorithms using only contractive compressors for distributed optimization in a heterogeneous setting. We omit the comparison in the general convex regime since most of the works focus on strongly convex and general nonconvex settings. **nCVX** = supports **nonconvex** functions; **sCVX** = supports **strongly convex** functions. We present the convergence $\mathbb{E}\left[\|\nabla f(\mathbf{x}_{\text{out}})\|^2\right] \leq \varepsilon$ in the nonconvex and $\mathbb{E}\left[f(\mathbf{x}_{\text{out}}) - f^\star\right]$ in the strongly convex regimes for specifically chosen $\mathbf{x}_{\text{out}}$. Here $F_0 := f(\mathbf{x}_0) - f^\star$.

| Algorithm | nCVX | sCVX |
|---|---|---|
| EC 
 Seide et al. (2014) | $\frac{LF_0\sigma^2}{n\varepsilon^2} + \frac{LF_0(\sigma+\zeta/\sqrt{\delta})}{\sqrt{\delta}\varepsilon^{3/2}} + \frac{LF_0}{\delta\varepsilon}$ (a) | $\frac{\sigma^2}{\mu n\varepsilon} + \frac{\sqrt{L}(\sigma+\zeta/\sqrt{\delta})}{\mu\sqrt{\delta\varepsilon}} + \frac{L}{\mu\delta}$ (a) |
| Choco-SGD 
 Koloskova et al. (2020a) | $\frac{L_{\max}F_0\sigma^2}{n\varepsilon^2} + \frac{L_{\max}F_0G}{\delta\varepsilon^{3/2}} + \frac{L_{\max}F_0}{\delta\varepsilon}$ (b) | $\frac{\sigma^2}{\mu n\varepsilon} + \frac{\sqrt{L_{\max}}G}{\mu\delta\varepsilon^{1/2}} + \frac{G^{2/3}}{\mu^{1/3}\delta\varepsilon^{1/3}}$ (b) |
| EF21-SGD 
 (Fatkhullin et al., 2021) | $\frac{\widetilde{L}F_0\sigma^2}{\delta^3\varepsilon^2} + \frac{\widetilde{L}F_0}{\delta\varepsilon}$ (c) | $\frac{\widetilde{L}\sigma^2}{\mu^2\delta^3\varepsilon} + \frac{\widetilde{L}}{\mu\delta}$ (c) |
| EF21-SGD2M 
 (Fatkhullin et al., 2023) | $\frac{LF_0\sigma^2}{n\varepsilon^2} + \frac{LF_0\sigma^{2/3}}{\delta^{2/3}\varepsilon^{4/3}} + \frac{\widetilde{L}F_0+\sigma\sqrt{LF_0}}{\delta\varepsilon}$ (d) | ✗ |
| EControl 
 **This work** | $\frac{LF_0\sigma^2}{n\varepsilon^2} + \frac{LF_0\sigma}{\delta^2\varepsilon^{3/2}} + \frac{\widetilde{L}F_0}{\delta\varepsilon}$ | $\frac{\sigma^2}{\mu n\varepsilon} + \frac{\sqrt{L}\sigma}{\mu\delta^2\varepsilon^{1/2}} + \frac{\widetilde{L}}{\mu\delta}$ |
| Lower Bounds 
 (He et al., 2023) | $\frac{LF_0\sigma^2}{n\varepsilon^2} + \frac{LF_0}{\delta\varepsilon}$ (e) | $\frac{\sigma^2}{\mu n\varepsilon} + \frac{1}{\delta}\sqrt{\frac{L}{\mu}}$ (f) |

(a) The analysis assumes a gradient dissimilarity bound of local gradients $\frac{1}{n}\sum_{i=1}^n \|\nabla f_i(\mathbf{x})\|^2 \leq \zeta^2 + \|\nabla f(\mathbf{x})\|^2$ (Stich, 2020).

(b) The analysis is done under the second moment bound of the stochastic gradients $\mathbb{E}\left[\|\mathbf{g}^i(\mathbf{x})\|^2\right] \leq G^2$. We emphasize that strongly convex functions do not satisfy this assumption.

(c) This result requires the assumption that *each* batch size is at least $\frac{\sigma^2}{\delta^2\varepsilon}$ in nonconvex regime and $\frac{\sigma^2}{\mu\delta^2\varepsilon}$ in the strongly convex regime.

(d) The last term becomes $\frac{\widetilde{L}F_0}{\delta\varepsilon}$ if the initial batch size is at least $\frac{\sigma^2}{LF_0}$.

(e) The analysis is done under gradient disimilarity assumption $\frac{1}{n}\sum_{i=1}^n \|\nabla f_i(\mathbf{x}) - \nabla f(\mathbf{x})\|^2 \leq \zeta^2$. Moreover, the result requires large mini-batches and performing $1/\delta$ communication rounds per iteration.

(f) The result requires large mini-batches and performing $1/\delta$ communication rounds per iteration. Moreover, the optimal parameters are chosen with an assumption that $\frac{1}{n}\sum_{i=1}^n \|\nabla f_i(\mathbf{x}^\star)\|^2 = b^2$ is known.

## 3 PROBLEM FORMULATION AND ASSUMPTIONS

We consider the distributed optimization problem

$$f^\star := \min_{\mathbf{x}\in\mathbb{R}^d}\left[f(\mathbf{x}) := \frac{1}{n}\sum_{i=1}^n f_i(\mathbf{x})\right], \tag{2}$$

where $\mathbf{x}$ represents the parameters of a model we aim to train, and the objective function $f : \mathbb{R}^d \to \mathbb{R}$ is decomposed into $n$ terms $f_i : \mathbb{R}^d \to \mathbb{R}, i \in [n] := \{1, \dots, n\}$. Each individual function $f_i(\mathbf{x}) := \mathbb{E}_{\xi_i\sim\mathcal{D}_i}[f_i(\mathbf{x}, \xi_i)]$ is a loss associated with a local dataset $\mathcal{D}_i$ available to client $i$. We let $\mathbf{x}^\star := \arg\min_{\mathbf{x}\in\mathbb{R}^d} f(\mathbf{x})$.

We analyze the centralized setting where the clients are connected with each other through a server. Typically, the server receives the messages from the clients and transfers back the aggregated information. In contrast to many prior works, we study arbitrarily heterogeneous setting and do not make any assumptions on the heterogeneity level, i.e. local data distributions might be far away from each other. We now list the main assumptions on the optimization problem (2).

First, we introduce assumptions on $f$ and $f_i$ that we use to derive convergence guarantees.

**Assumption 1.** *We assume that the average function $f$ has $L$-Lipschitz gradient, and each individual function $f_i$ has $L_i$-Lipschitz gradient, i.e. for all $\mathbf{x}, \mathbf{y} \in \mathbb{R}^d$, and $i \in [n]$ it holds*

$$\|\nabla f(\mathbf{y}) - \nabla f(\mathbf{x})\| \leq L\|\mathbf{y} - \mathbf{x}\|, \quad \|\nabla f_i(\mathbf{y}) - \nabla f_i(\mathbf{x})\| \leq L_i\|\mathbf{y} - \mathbf{x}\|. \tag{3}$$

We define a constant $\widetilde{L} := \sqrt{\frac{1}{n}\sum_{i=1}^n L_i^2}$.[2] We note that most works derive convergence guarantees with maximum smoothness constant $L_{\max} := \max_{i\in[n]} L_i$ which can be much larger than $\widetilde{L}$. Next, in some cases we assume $\mu$-strong convexity of $f$.

---

[2] Note that the following chain of inequalities always hold: $L \leq \widetilde{L} \leq L_{\max} := \max_{i\in[n]} L_i$.

| **Algorithm 1** EC-Ideal | **Algorithm 2** EControl |
|---|---|
| 1: **Input:** $\mathbf{x}_0, \mathbf{e}_t^i = \mathbf{0}_d, \mathbf{h}_\star^i := \nabla f_i(\mathbf{x}^\star), \gamma, \mathcal{C}_\delta$ | 1: **Input:** $\mathbf{x}_0, \mathbf{e}_0^i = \mathbf{0}_d, \mathbf{h}_0^i = \mathbf{g}_0^i, \gamma, \eta,$ and $\mathcal{C}_\delta$ |
| 2: $\mathbf{h}_\star = \frac{1}{n}\sum_{i=1}^n \mathbf{h}_\star^i$ | 2: $\mathbf{h}_0 = \frac{1}{n}\sum_{i=1}^n \mathbf{h}_0^i$ |
| 3: **for** $t = 0, 1, 2, \ldots$ **do** | 3: **for** $t = 0, 1, 2, \ldots$ **do** |
| 4:    **client side:** | 4:    **client side:** |
| 5:    compute $\mathbf{g}_t^i = \mathbf{g}^i(\mathbf{x}_t)$ | 5:    compute $\mathbf{g}_t^i = \mathbf{g}^i(\mathbf{x}_t)$ |
| 6:    compute $\Delta_t^i = \mathcal{C}_\delta(\mathbf{e}_t^i + \mathbf{g}_t^i - \mathbf{h}_\star^i)$ | 6:    compute $\Delta_t^i = \mathcal{C}_\delta(\eta\mathbf{e}_t^i + \mathbf{g}_t^i - \mathbf{h}_t^i)$ |
| 7:    update $\mathbf{e}_{t+1}^i = \mathbf{e}_t^i + \mathbf{g}_t^i - \mathbf{h}_\star^i - \Delta_t^i$ | 7:    update $\mathbf{e}_{t+1}^i = \mathbf{e}_t^i + \mathbf{g}_t^i - \mathbf{h}_t^i - \Delta_t^i$ |
| 8:    send to server $\Delta_t^i$ | 8:    and $\mathbf{h}_{t+1}^i = \mathbf{h}_t^i + \Delta_t^i$ |
| 9:    **server side:** | 9:    send to server $\Delta_t^i$ |
| 10:    update $\mathbf{x}_{t+1} := \mathbf{x}_t - \gamma\mathbf{h}_\star - \frac{\gamma}{n}\sum_{i=1}^n \Delta_t^i$ | 10:   **server side:** |
| | 11:    update $\mathbf{x}_{t+1} := \mathbf{x}_t - \gamma\mathbf{h}_t - \frac{\gamma}{n}\sum_{i=1}^n \Delta_t^i$ |
| | 12:    and $\mathbf{h}_{t+1} = \mathbf{h}_t + \frac{1}{n}\sum_{i=1}^n \Delta_t^i$ |

**Assumption 2.** *We assume that the average function $f$ is $\mu$-strongly convex, i.e. for all $\mathbf{x}, \mathbf{y} \in \mathbb{R}^d$ it holds*

$$f(\mathbf{x}) \geq f(\mathbf{y}) + \langle \nabla f(\mathbf{y}), \mathbf{x} - \mathbf{y} \rangle + \frac{\mu}{2}\|\mathbf{x} - \mathbf{y}\|^2. \tag{4}$$

*We say that $f$ is convex if it satisfies (4) with $\mu = 0$.*

We highlight that we need the strong convexity (convexity resp.) only of the average function $f$ while individual functions $f_i$ do not have to satisfy it, and they can be even nonconvex. On top of that, in our analysis we apply the strong convexity (convexity resp.) around $\mathbf{x}^\star$ only, thus, it is sufficient to assume that (4) holds for any $\mathbf{y}$ and $\mathbf{x} \equiv \mathbf{x}^\star$. In this case, we say that a function is $\mu$-strongly quasi-convex (quasi-convex resp.) around $\mathbf{x}^\star$; see (Stich & Karimireddy, 2020).

Finally, we make an assumption on the noise of local gradient estimators used by the clients.

**Assumption 3.** *We assume that we have access to a gradient oracle $\mathbf{g}^i(\mathbf{x}): \mathbb{R}^d \to \mathbb{R}^d$ for each local function $f_i$ such that for all $\mathbf{x} \in \mathbb{R}^d$ and $i \in [n]$ it holds*

$$\mathbb{E}\left[\mathbf{g}^i(\mathbf{x})\right] = \nabla f_i(\mathbf{x}), \quad \mathbb{E}\left[\|\mathbf{g}^i(\mathbf{x}) - \nabla f_i(\mathbf{x})\|^2\right] \leq \sigma^2. \tag{5}$$

Note that mini-batches are also allowed, effectively dividing this variance quantity by the local batch size. However, we do not need to impose any restrictions on the (minimal) batch size and, for simplicity, always assume a batch size of one.

## 4 EC-Ideal AS A STARTING POINT

To motivate some of the key ingredients of our main method, we start with a simple idea developed in an ideal setting, which shed some light on the nuances of the interactions between bias and error correction. Assume for a moment that we have access to $\mathbf{h}_\star^i := \nabla f_i(\mathbf{x}^\star)$. We can utilize this additional information to modify the original EC mechanism so that we only compress the difference between the current stochastic gradient $\mathbf{g}_t^i$ and the $\mathbf{h}_\star^i$. See Algorithm 1 and the highlight below:

$$\text{EC-Ideal update:} \quad \begin{aligned} \Delta_t^i &= \mathcal{C}_\delta(\mathbf{e}_t^i + \mathbf{g}_t^i - \mathbf{h}_\star^i) \\ \mathbf{e}_{t+1}^i &= \mathbf{e}_t^i + \mathbf{g}_t^i - \mathbf{h}_\star^i - \Delta_t^i. \end{aligned} \tag{6}$$

It turns out that this simple trick leads to a dramatic theoretical improvement. Now we do not have to restrict the heterogeneity of the problem in contrast to the analysis of original EC (Stich, 2020). The next theorem presents the convergence of EC-Ideal.

**Theorem 1** (Convergence of EC-Ideal). *Let $f : \mathbb{R}^d \to \mathbb{R}$ be $\mu$-strongly quasi-convex around $\mathbf{x}^\star$, and each $f_i$ be L-smooth and convex. Then there exists a stepsize $\gamma \leq \frac{\delta}{8\sqrt{6}L}$ such that after at most*

$$T = \widetilde{\mathcal{O}}\left(\frac{\sigma^2}{\mu n\varepsilon} + \frac{\sqrt{L}\sigma}{\mu\sqrt{\delta}\varepsilon^{1/2}} + \frac{L}{\mu\delta}\right)$$

*iterations of Algorithm 1 it holds* $\mathbb{E}[f(\mathbf{x}_{out}) - f^\star] \leq \varepsilon$, *where* $\mathbf{x}_{out}$ *is chosen randomly from* $\{\mathbf{x}_0, \ldots, \mathbf{x}_T\}$ *with probabilities proportional to* $(1 - \mu\gamma/2)^{-(t+1)}$.

We defer the proof to Appendix E. Note that the output criteria (selecting a random iterate) is standard in the literature and for convex functions could also be replaced by a weighted average over all iterates that can be efficiently tracked over time (see e.g. Rakhlin et al., 2011). EC-Ideal achieves very desirable properties. First, it provably works for any contractive compression. Second, it achieves optimal asymptotic complexity and the standard linear convergence when $\sigma^2 \to 0$. All of these results are derived without enforcing data heterogeneity bounds. However, there are several drawbacks as well. First, EC-Ideal requires knowledge of $\{\mathbf{h}_\star^i\}_{i \in [n]}$ which is unrealistic in most applications. In Appendix D we show that the algorithm, in fact, needs only approximations of $\{\mathbf{h}_\star^i\}_{i \in [n]}$ as input. Nonetheless, there is still an issue that EC-Ideal's convergence is only guaranteed under the convexity condition, which stems from the fact that we always need to control the distance between $\nabla f_i(\mathbf{x}_t)$ and $\nabla f_i(\mathbf{x}^\star)$. To overcome this issue, we need to build estimators of the current gradient, instead of the gradient at the optimum, parallel to maintaining the error term.

## 5    EControl IS A SOLUTION TO ALL ISSUES

Inspired by the properties of EC-Ideal, we aim to build a mechanism that will progressively learn the local gradient approximations (see also Remark 15 in Appendix B). Stich (2020); Gorbunov et al. (2020) created a learning process based on an additional compressor from a more restricted class of unbiased compression operators. To make their method work for any contractive compressor, we can replace the additional unbiased compressor with a more general contractive one, but this leads to worse complexity in the low noise regime and more restriction on the stepsize. We refer the reader to Appendix F for more detailed discussion. In this work, we propose a novel method, called EControl, that overcomes all of aforementioned issues by

- building the error term and the gradient estimator with a single compressed message, enabling error compensation for the gradient estimator;

- introducing a new parameter $\eta$ to precisely *control* the error compensation and balance error compensation carried from the gradient estimator.

We summarize EControl in Algorithm 2 and highlight the key ingredients as follows:

$$\text{EControl update:} \quad \begin{aligned} \Delta_t^i &= \mathcal{C}_\delta(\eta \mathbf{e}_t^i + \mathbf{g}_t^i - \mathbf{h}_t^i) \\ \mathbf{e}_{t+1}^i &= \mathbf{e}_t^i + \mathbf{g}_t^i - \mathbf{h}_t^i - \Delta_t^i \\ \mathbf{h}_{t+1}^i &= \mathbf{h}_t^i + \Delta_t^i \end{aligned} \qquad (7)$$

Fusing the error and the gradient estimator updates improves the algorithm's dependency on $\sigma^2$ and $\delta$, but also enables the gradient estimator to carry some level of the error information. This brings forth the challenge of balancing the strength of the feedback signal, and we introduce the $\eta$ parameter to obtain finer control over the error compensation mechanism. This new parameter brings additional flexibility and allows us to stabilize the convergence of EControl. The effect of such a parameter in the context of the original EC mechanism without gradient estimator might be of independent interest. We observe in practice that Algorithm 2 with $\eta = 1$ is unstable and sensitive to the choice of initialization, which we illustrate in Appendix C.2 with a toy example. This highlights the importance of the finer control over EC that $\eta$ enables. We refer the interested readers to Appendix C.1 for a more detailed discussion on the importance of $\eta$ from a theoretical perspective. In Appendix C.3 we also discuss the connection between EControl and EF21 when $\eta \to 0$. A similar idea of weighting error terms appeared in Abdi & Fekri (2020); Li et al. (2023). However, their algorithm is based on the original EC.

## 6    THEORETICAL ANALYSIS OF EControl

We move on to theoretical analysis of EControl. Below we summarize the convergence properties of Algorithm 2 in all standard cases. We start with the convergence results in the strongly quasi-convex regime.

**Theorem 2** (Convergence of EControl for strongly quasi-convex objective). *Let $f$ be $\mu$-strongly quasi-convex around $\mathbf{x}^\star$ and $L$-smooth. Let each $f_i$ be $L_i$-smooth. Then for $\eta = c\delta$ (where $c$ is an absolute constant we specify in the proof), there exists a $\gamma \leq \mathcal{O}(\delta/\widetilde{L})^3$. such that after at most*

$$T = \widetilde{\mathcal{O}}\left(\frac{\sigma^2}{\mu n \varepsilon} + \frac{\sqrt{L}\sigma}{\mu \delta^2 \varepsilon^{1/2}} + \frac{\widetilde{L}}{\mu \delta}\right)$$

*iterations of Algorithm 2 it holds $\mathbb{E}\left[f(\mathbf{x}_{out}) - f^\star\right] \leq \varepsilon$, where $\mathbf{x}_{out}$ is chosen randomly from $\mathbf{x}_t \in \{\mathbf{x}_0, \ldots, \mathbf{x}_T\}$ with probabilities proportional to $(1 - \frac{\gamma\mu}{2})^{-(t+1)}$.*

The asymptotic complexity of EControl in this regime with stochastic gradient matches the lower bound $\Omega(\frac{\sigma^2}{\mu n \varepsilon})$ up to a log term, as opposed to previous works (Fatkhullin et al., 2021; Zhao et al., 2022). As we can see, the first term linearly improves with $n$, similar to the convergence behavior of distributed SGD. Moreover, EControl achieves standard linear convergence and a desired inverse linear dependency on the compression parameter $\delta$ in the noiseless setting. Next, we switch to a general convex setting.

**Theorem 3** (Convergence of EControl for quasi-convex objective). *Let $f$ be quasi-convex around $\mathbf{x}^\star$ and $L$-smooth. Let each $f_i$ be $L_i$-smooth. Then for $\eta = c\delta$ there exists a $\gamma \leq \mathcal{O}(\delta/\widetilde{L})$ such that after at most*

$$T = \mathcal{O}\left(\frac{R_0\sigma^2}{n\varepsilon^2} + \frac{\sqrt{L}R_0\sigma}{\delta^2\varepsilon^{3/2}} + \frac{\widetilde{L}R_0}{\delta\varepsilon}\right)$$

*iterations of Algorithm 2 it holds $\mathbb{E}\left[f(\mathbf{x}_{out}) - f^\star\right] \leq \varepsilon$, where $R_0 := \|\mathbf{x}_0 - \mathbf{x}^\star\|^2$ and $\mathbf{x}_{out}$ is chosen uniformly at random from $\mathbf{x}_t \in \{\mathbf{x}_0, \ldots, \mathbf{x}_T\}$.*

Similar to the previous case, the first term enjoys a linear speedup by the number of nodes. We achieve a standard sublinear convergence and a desired inverse linear dependency on $\delta$ in the noiseless regime.

**Theorem 4** (Convergence of EControl for nonconvex objective). *Let $f$ be $L$-smooth, and each $f_i$ be $L_i$-smooth. Then for $\eta = c\delta$ there exists a $\gamma \leq \mathcal{O}(\delta/\widetilde{L})$ such that after at most*

$$T = \mathcal{O}\left(\frac{LF_0\sigma^2}{n\varepsilon^2} + \frac{LF_0\sigma}{\delta^2\varepsilon^{3/2}} + \frac{\widetilde{L}F_0}{\delta\varepsilon}\right)$$

*iterations of Algorithm 2 it holds $\mathbb{E}\left[\|\nabla f(\mathbf{x}_{out})\|^2\right] \leq \varepsilon$, where $F_0 := f(\mathbf{x}_0) - f^\star$ and $\mathbf{x}_{out}$ is chosen uniformly at random from $\mathbf{x}_t \in \{\mathbf{x}_0, \ldots, \mathbf{x}_T\}$.*

The complexity of EControl in both asymptotic and noiseless regimes matches known lower bounds (He et al., 2023) for nonconvex functions that are derived under much stronger assumptions. We achieve these results without any theoretical restrictions, such as large batch sizes or repeated communication rounds, making it easy to deploy in practice as an extension to existing EC implementations.

# 7 EXPERIMENTS

In this section, we complement our theoretical analysis of EControl with experimental results. We corroborate our theoretical findings with experimental evaluations and illustrate the practical efficiency and effectiveness of EControl in real-world scenarios.

## 7.1 SYNTHETIC LEAST SQUARES PROBLEM

First, we consider the least squares problem designed by Koloskova et al. (2020b). For each client $i$, $f_i(\mathbf{x}) := \frac{1}{2}\|\mathbf{A}_i\mathbf{x} - \mathbf{b}_i\|^2$, where $\mathbf{A}_i^2 := \frac{i^2}{n} \cdot \mathbf{I}_d$ and each $\mathbf{b}_i$ is sampled from $\mathcal{N}(0, \frac{\zeta^2}{i^2}\mathbf{I}_d)$ for some parameter $\zeta$. The parameter $\zeta$ controls the gradient dissimilarity of the problem. It's easy to see that

---

[3]Note that the stepsize can depend on $\varepsilon$. Here we use the notation $\gamma \leq \mathcal{O}(\delta/\widetilde{L})$ to denote that the stepsize must satisfy $\gamma \leq \frac{C\delta}{\widetilde{L}}$, for an absolute constant $C$ specified in the proof.

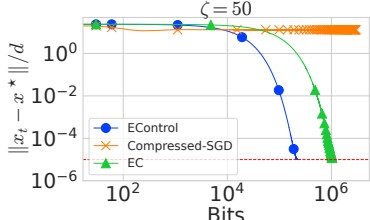
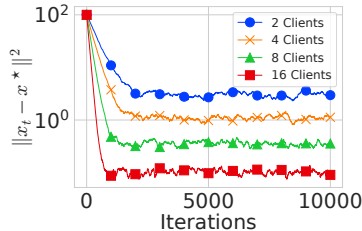

Figure 1: The comparison of Compressed-SGD, EC, and EControl. $n = 5, d = 300, \zeta = 50$ and $\sigma = 10$. We apply Top-K compressor with $K/d = 0.1$. Stepsizes were tuned for each setting. X-axis represents the number of bits sent. Compressed-SGD (without error compensation) does not converge and EControl slightly outperforms the classic EC.

Figure 2: The behavior of EControl with different numbers of clients where $d = 200, \zeta = 100, \sigma = 50$. We apply Top-K compressor with $K/d = 0.1$. The stepsize is fixed $\gamma = \frac{\delta}{100} = 0.001$ for the purpose of illustration. EControl exhibits a clear linear speedup by $n$, the number of clients, which verifies the linear parallel speedup property of EControl.

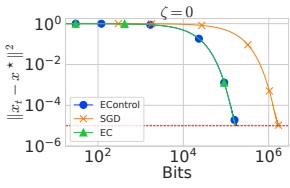
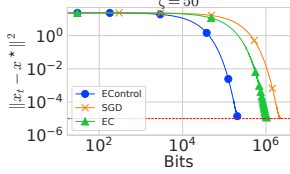
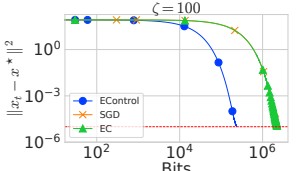

Figure 3: Comparison of mini-batch SGD, EC, and EControl for vairous problem parameters where $n = 5, d = 300$ and $\sigma = 10$. We apply the Top-K compressor with $K/d = 0.1$. Stepsizes were tuned for each setting. X-axis represents the number of bits sent. EControl is not affected by the data heterogeneity parameter and is superior over SGD in terms of the number of bits sent (roughly by a factor of $d/K = 10$, as expected from theory).

when $\zeta = 0, \nabla f_i(\mathbf{x}^\star) = 0, \forall i$. We add Gaussian noise to the gradients to control the stochastic level $\sigma^2$ of the gradient. In Figure 1 one can see that simply aggregating the compressed local gradient without error compensation (termed Compressed-SGD (Khirirat et al., 2018; Alistarh et al., 2018)) does not lead to convergence, while EControl and EC do. This result shows the need to use EC to make the method convergent. Further, we use this simple synthetic problem to demonstrate some of the key features of EControl:

**Increasing the number of clients.** In Figure 2 we investigate the effect of the number of clients on the complexity of EControl. Crucially, the theory predicts that EControl achieves a linear speedup in terms of the number of clients when using a stochastic gradient. In our experiment, we fix a small stepsize and investigate the error that EControl converges to. We see that as the number of clients doubles, the error that EControl oscillates around is roughly divided by half. This confirms our theoretical prediction of the linear speedup.

**Independence from gradient dissimilarity.** In Figure 3 we see that EControl is not affected by the gradient dissimilarity parameter $\zeta$ and its complexity stays stable across the three figures. On the other hand, the original EC suffers from the increasing $\zeta$, taking longer to converge as $\zeta$ increases. Moreover, as Theorem 2 predicts, the $\mathcal{O}(\frac{\sigma^2}{\mu n \varepsilon})$ term is dominant, and the performance of EControl (in terms of the number of bits sent) is superior over that of SGD with batch size $n$. The stepsizes $\gamma$ are fine-tuned over $\{5 \times 10^{-5}, 10^{-4}, 5 \times 10^{-4}, 10^{-3}, 10^{-2}, 10^{-1}\}$, and for EControl we fine-tune $\eta$ over $\{10^{-3}, 5 \times 10^{-3}, 10^{-2}, 5 \times 10^{-2}, 10^{-1}\}$. As $\zeta$ increases, the performance of EControl is not affected, while the performance of EC deteriorates.

## 7.2  LOGISTIC REGRESSION PROBLEM

Next, we consider the Logistic Regression problem for multi-class classification trained on MNIST dataset (Deng, 2012) and implemented in Pytorch (Paszke et al., 2019). First, we split $50\%$ of the dataset between 10 clients according to the labels (the data point with $i$-th labels belongs to client $i + 1$). The rest of the data is distributed randomly between clients. Then, for each client, we divide the local data into train ($90\%$) and test ($10\%$) sets. Such partition allows to make the problem

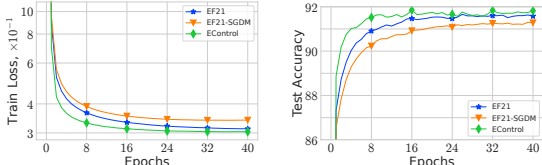

Figure 4: The comparison of EF21, EF21-SGDM, and EControl with fine-tuned parameters for Logistic Regression problem on MNIST dataset.

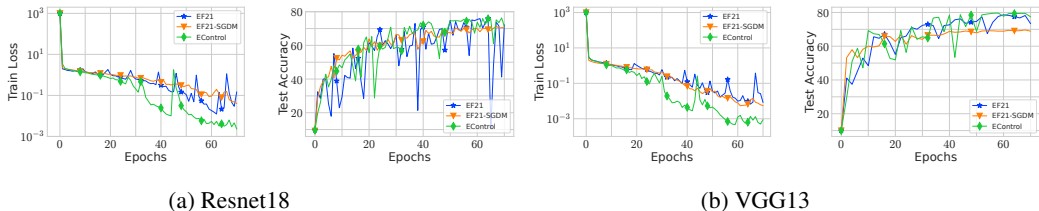

(a) Resnet18                    (b) VGG13

Figure 5: The comparison of EF21, EF21-SGDM, and EControl with fine-tuned parameters for training Deep Learning models on Cifar-10 dataset.

more heterogeneous. We compare the performance of EControl, EF21-SGDM, and EF21. For all methods, we use Top-K compression operator with $K = \frac{d}{10}$. We fine-tune the stepsizes of the methods over $\{1, 10^{-1}, 10^{-2}, 10^{-3}\}$. Moreover, we fine-tune $\eta$ parameter for EControl over $\{0.2, 0.1, 0.05\}$, and for EF21-SGDM we set $\eta = 0.1$ according to (Fatkhullin et al., 2023). We choose the stepsizes that achieve the best performances on the train set such that the test loss does not diverge. The results are presented in Figure 4. Note that the communication cost of the methods per iteration is the same, therefore, the number of epochs is proportional to the number of communicated bits.

We observe that EControl has the fastest convergence in terms of training loss. The same behavior is demonstrated with respect to test accuracy. Nevertheless, the test accuracy difference for all methods is within 1 percent.

### 7.3 TRAINING OF DEEP LEARNING MODELS

Finally, we consider the training of Deep Learning models: Resnet18 (He et al., 2016) and VGG13 (Simonyan & Zisserman, 2015). The implementation is done in Pytorch (Paszke et al., 2019). We run experiments on Cifar-10 (Krizhevsky et al., 2014) dataset. The dataset split across the clients is the same as for the Logistic Regression problem—half is distributed randomly, and another half is portioned according to the labels. For the compression, we utilize Top-K compressor with $\frac{K}{d} = 0.1$. We fine-tune the stepsizes over the set $\{1, 0.1, 0.01\}$ for EControl, EF21-SGDM, and EF21 and select the best stepsize on test set. For EControl and EF21-SGDM we set $\eta = 0.1$.

According to the results in Figure 5a, all methods achieve similar test accuracy, but EF21 has more unstable convergence with many ups and downs. EControl achieves a better stationary point as the training loss is smaller than for the other two methods. Similar results are demonstrated for VGG13 model; see Figure 5b. The training with EControl allows to reach a stationary point with a smaller training loss. Moreover, EControl outperforms other methods from a test accuracy point of view: it gives slightly better test accuracy than EF21 and considerably higher accuracy than EF21-SGDM.

## 8 DISCUSSION

In this work, we propose a novel algorithm EControl that provably converges efficiently in all standard settings: strongly convex, general convex, and nonconvex, with general contractive compression and without any additional assumption on the problem structure, hence resolving the open theoretical problem therein. We conduct extensive experimental evaluations of our method and show its efficacy in practice. Our method incurs little overhead compared to existing implementations of EC, and we believe it is an effective and lightweight approach to making EC suitable for distributed training of large machine learning models, especially when also using the standard momentum mechanism for reducing the variance of the stochastic gradients in deep learning training. However, the theoretical analysis of this extension is outside the scope of this paper, and we will leave it for future work.

## ACKNOWLEDGMENTS

YG/SS acknowledge partial funding from a Google Scholar Research Award.

## REPRODUCIBILITY STATEMENT

We provide source code as part of the supplementary material which allows to reproduce Deep Learning and synthetic least squares experiments. Additionally, for our theoretical results, we offer clear explanations of any assumptions in the main paper and complete proof of all statements in the appendix.

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

## A  USEFUL LEMMAS AND DEFINITIONS

In this section, we state the important notations and useful lemmas that we use in our convergence analysis. We use the following notation throughout the proofs

$$F_t := \mathbb{E}\left[f(\mathbf{x}_t)\right] - f^\star, \quad \text{and} \quad E_t := \frac{1}{n}\sum_{i=1}^{n}\mathbb{E}\left[\left\|\mathbf{e}_t^i\right\|^2\right] \tag{8}$$

For shortness, in all our proofs we use the notation

$$\mathbf{g}_t^i := \mathbf{g}^i(\mathbf{x}_t), \quad \mathbf{g}_t := \frac{1}{n}\sum_{i=1}^{n}\mathbf{g}_t^i, \quad \mathbf{e}_t := \frac{1}{n}\sum_{i=1}^{n}\mathbf{e}_t^i, \quad \text{and} \quad \mathbf{h}_t := \frac{1}{n}\sum_{i=1}^{n}\mathbf{h}_t^i. \tag{9}$$

Besides, we additionally introduce the sequence of virtual iterates which are defined as follows

$$\widetilde{\mathbf{x}}_0 := \mathbf{x}_0, \quad \widetilde{\mathbf{x}}_{t+1} := \widetilde{\mathbf{x}}_t - \frac{\gamma}{n}\sum_{i=1}^{n}\mathbf{g}_t^i. \tag{10}$$

Performing simple derivations we get the link between real and virtual iterates of Algorithm 2

$$\mathbf{x}_t - \widetilde{\mathbf{x}}_t = \frac{\gamma}{n}\sum_{i=1}^{n}\mathbf{e}_t^i. \tag{11}$$

In addition to the notations introduced inEquation (8), we define

$$X_t := \mathbb{E}\left[\|\widetilde{\mathbf{x}}_t - \mathbf{x}^\star\|^2\right]. \tag{12}$$

Finally, we introduce another quantity that bounds the size of the compressed message:

$$H_t := \frac{1}{n}\sum_{i=1}^{n}\mathbb{E}\left[\left\|\eta\mathbf{e}_t^i + \mathbf{g}_t^i - \mathbf{h}_t^i\right\|^2\right] \tag{13}$$

The following lemmas were taken from other works (as indicated) and we omit their proof.

**Lemma 1** (Lemma 8 from Stich & Karimireddy (2020)). *Assume $f$ is $L$-smooth and $\mu$-strongly quasi-convex. Let the sequences $\{\mathbf{x}_t\}$, $\{\mathbf{e}_t^i\}_{i\in[n]}$ be generated by Algorithm 2. If $\gamma \le \frac{1}{4L}$, then*

$$X_{t+1} \le \left(1 - \frac{\gamma\mu}{2}\right)X_t - \frac{\gamma}{2}F_t + \frac{\gamma^2}{n}\sigma^2 + 3L\gamma^3 E_t. \tag{14}$$

The descent lemma in $\widetilde{F}_t$ is taken from (Stich & Karimireddy, 2020)

**Lemma 2** (Lemma 9 from Stich & Karimireddy (2020)). *Assuming that $f$ is $L$-smooth, and $\mathbf{x}_t$ and $\mathbf{e}_t$ are generated by Algorithm 2, then*

$$\widetilde{F}_{t+1} \le \widetilde{F}_t - \frac{\gamma}{4}\mathbb{E}\left[\|\nabla f(\mathbf{x}_t)\|^2\right] + \frac{\gamma^3 L^2}{2}E_t + \frac{\gamma^2 L\sigma^2}{2n} \tag{15}$$

For the sake of completeness, we list two summation lemmas from (Stich, 2020) without proofs. The first lemma is used in order to derive convergence guarantees in the strongly convex case.

**Lemma 3** (Lemma 25 from Stich (2020)). *Let $\{r_t\}_{t\ge 0}$ and $\{s_t\}_{t\ge 0}$ be sequences of positive numbers satisfying*

$$r_{t+1} \le (1 - \min\{\gamma A, F\})r_t - B\gamma s_t + C\gamma^2 + D\gamma^3, \tag{16}$$

*for some positive constants $A, B > 0$, $C, D \ge 0$, and for constant stepsize $0 < \gamma \le \frac{1}{E}$, for $E \ge 0$, and for parameter $0 < F \le 1$. Then there exists a constant stepsize $\gamma \le \frac{1}{E}$ such that*

$$\frac{B}{W_T}\sum_{t=0}^{T}w_t s_t + \min\left\{A, \frac{F}{\gamma}\right\}r_{T+1} \le r_0\left(E + \frac{A}{F}\right)\exp\left(-\min\left\{\frac{A}{E}, F\right\}(T+1)\right)$$

$$+ \frac{2C\ln\tau}{A(T+1)} + \frac{D\ln^2\tau}{A^2(T+1)^2}$$

*for $w_t := (1 - \min\{\gamma A, F\})^{-(t+1)}$, $W_T := \sum_{t=0}^{T}w_t$, and*

$$\tau := \max\left\{\exp(1), \min\left\{\frac{A^2 r_0(T+1)^2}{C}, \frac{A^3 r_0(T+1)^3}{D}\right\}\right\}.$$

**Remark 4** (Remark 26 from Stich (2020)). *Lemma 3 established a bound of the order*

$$\widetilde{\mathcal{O}}\left(r_0\left(E + \frac{A}{F}\right)\exp\left(-\min\left\{\frac{A}{E}, F\right\}(T+1)\right) + \frac{C}{AT} + \frac{D}{A^2T^2}\right)$$

*that decreases with $T$. To ensure that this expression is smaller than $\varepsilon$,*

$$T = \widetilde{\mathcal{O}}\left(\frac{C}{A\varepsilon} + \frac{\sqrt{D}}{A\sqrt{\varepsilon}} + \frac{1}{F}\log\frac{1}{\varepsilon} + \frac{E}{A}\log\frac{1}{\varepsilon}\right) = \widetilde{\mathcal{O}}\left(\frac{C}{A\varepsilon} + \frac{\sqrt{D}}{A\sqrt{\varepsilon}} + \frac{1}{F} + \frac{E}{A}\right)$$

*steps are sufficient.*

The second lemma shows the convergence rate in the nonconvex or convex settings.

**Lemma 5** (Lemma 27 from Stich (2020)). *Let $\{r_t\}_{t\geq 0}$ and $\{s_t\}_{t\geq 0}$ be sequences of positive numbers satisfying*

$$r_{t+1} \leq r_t - B\gamma s_t + C\gamma^2 + D\gamma^3, \tag{17}$$

*for some positive constants $A, B > 0, C, D \geq 0$, and for constant stepsize $0 < \gamma \leq \frac{1}{E}$, for $E \geq 0$, and for parameter $0 < F \leq 1$. Then there exists a constant stepsize $\gamma \leq \frac{1}{E}$ such that*

$$\frac{B}{T+1}\sum_{t=0}^{T} s_t \leq \frac{Er_0}{T+1} + 2D^{1/3}\left(\frac{r_0}{T+1}\right)^{2/3} + 2\left(\frac{Cr_0}{T+1}\right)^{1/2}. \tag{18}$$

**Remark 6** (Remark 28 from Stich (2020)). *To ensure that the right-hand side of (18) is smaller than $\varepsilon > 0$,*

$$T = \mathcal{O}\left(\frac{Cr_0}{\varepsilon^2} + \frac{\sqrt{D}r_0}{\varepsilon^{3/2}} + \frac{Er_0}{\varepsilon}\right).$$

*steps are sufficient.*

# B   MISSING PROOFS FOR EControl

In this section we prove Theorem 2, Theorem 3 and Theorem 4.

## B.1   STRONGLY CONVEX SETTING

We start with the strongly convex case setting. First, we bound the distance between two consecutive iterates.

**Lemma 7.** *Let $f$ be $L$-smooth, then:*

$$\mathbb{E}\left[\|\mathbf{x}_{t+1} - \mathbf{x}_t\|^2\right] \leq \gamma^2\left(2(1-\delta)H_t + 4\eta^2 E_t + 4LF_t + \frac{2\sigma^2}{n}\right). \tag{19}$$

*Proof.*

$$
\begin{aligned}
\frac{1}{\gamma^2}\mathbb{E}\left[\|\mathbf{x}_{t+1} - \mathbf{x}_t\|^2\right] &= \mathbb{E}\left[\|\mathbf{h}_t + \Delta_t\|^2\right]\\
&= \mathbb{E}\left[\|\Delta_t - \eta\mathbf{e}_t - \mathbf{g}_t + \mathbf{h}_t + \eta\mathbf{e}_t + \mathbf{g}_t\|^2\right]\\
&\overset{(i)}{\leq} 2\mathbb{E}\left[\|\Delta_t - \eta\mathbf{e}_t - \mathbf{g}_t + \mathbf{h}_t\|^2\right] + 2\mathbb{E}\left[\|\eta\mathbf{e}_t + \mathbf{g}_t\|^2\right]\\
&\overset{(ii)}{\leq} 2\mathbb{E}\left[\|\Delta_t - \eta\mathbf{e}_t - \mathbf{g}_t + \mathbf{h}_t\|^2\right] + 2\mathbb{E}\left[\|\eta\mathbf{e}_t + \nabla f(\mathbf{x}_t)\|^2\right] + \frac{2\sigma^2}{n}\\
&\overset{(iii)}{\leq} \frac{2}{n}\sum_{i=1}^{n}\mathbb{E}\left[\|\Delta_t^i - \eta\mathbf{e}_t^i - \mathbf{g}_t^i + \mathbf{h}_t^i\|^2\right] + \frac{4\eta^2}{n}\sum_{i=1}^{n}\mathbb{E}\left[\|\mathbf{e}_t^i\|^2\right]\\
&\quad + 4\mathbb{E}\left[\|\nabla f(\mathbf{x}_t)\|^2\right] + \frac{2\sigma^2}{n}\\
&\overset{(iv)}{\leq} \frac{2(1-\delta)}{n}\sum_{i=1}^{n}\mathbb{E}\left[\|\eta\mathbf{e}_t^i + \mathbf{g}_t^i - \mathbf{h}_t^i\|^2\right] + \frac{4\eta^2}{n}\sum_{i=1}^{n}\mathbb{E}\left[\|\mathbf{e}_t^i\|^2\right]\\
&\quad + 4L\mathbb{E}\left[f(\mathbf{x}_t) - f^\star\right] + \frac{2\sigma^2}{n}.
\end{aligned}
$$

where in $(i) - (iii)$ we use Young's inequality; in $(ii)$ we use Assumption 5, and in $(iv)$ we use the definition of the compressor, $L$-smoothness and convexity. $\qquad\square$

The next lemma gives the descent of $E_t$.

**Lemma 8.** *For any $\alpha > 0$ we have:*

$$
E_{t+1} \leq (1+\alpha)(1-\eta)^2 E_t + (1+\alpha^{-1})(1-\delta)H_t \tag{20}
$$

*Proof.*

$$
\begin{aligned}
E_{t+1} &= \frac{1}{n}\sum_{i=1}^{n}\mathbb{E}\left[\|\mathbf{e}_t^i + \mathbf{g}_t^i - \mathbf{h}_t^i - \Delta_t^i\|^2\right]\\
&\overset{(i)}{\leq} (1+\alpha^{-1})\frac{1}{n}\sum_{i=1}^{n}\mathbb{E}\left[\|\eta\mathbf{e}_t^i + \mathbf{g}_t^i - \mathbf{h}_t^i - \Delta_t^i\|^2\right] + (1+\alpha)(1-\eta)^2 E_t\\
&\overset{(ii)}{\leq} (1+\alpha^{-1})(1-\delta)\frac{1}{n}\sum_{i=1}^{n}\mathbb{E}\left[\|\eta\mathbf{e}_t^i + \mathbf{g}_t^i - \mathbf{h}_t^i\|^2\right] + (1+\alpha)(1-\eta)^2 E_t\\
&= (1+\alpha^{-1})(1-\delta)H_t + (1+\alpha)(1-\eta)^2 E_t.
\end{aligned}
$$

where in $(i)$ we use Young's inequality, and in $(ii)$ we use the definition of the compressor. $\qquad\square$

Now we give the descent of $H_t$.

**Lemma 9.** *Let $f$ be $L$-smooth and each $f_i$ be $L_i$-smooth. Let $\eta = \frac{\delta}{4k}$ for some $k \geq 1$ and $\gamma \leq \frac{\delta}{32\sqrt{2}\widetilde{L}}$. Then we have:*

$$
\begin{aligned}
H_{t+1} &\leq \left(1 - \frac{\delta}{32}\right)H_t + \left(\frac{8\delta^3}{k^4 4^4} + \frac{128\widetilde{L}^2\gamma^2\delta}{k^2 4^2}\right)E_t + \frac{128\widetilde{L}^2 L\gamma^2}{\delta}F_t\\
&\quad + \frac{64}{\delta}\left(1 + \frac{\widetilde{L}^2\gamma^2}{n}\right)\sigma^2.
\end{aligned} \tag{21}
$$

*Proof.* We unroll the definition of $H_{t+1}$

$$
\begin{aligned}
H_{t+1} &= \frac{1}{n}\sum_{i=1}^{n}\mathbb{E}\left[\left\|\eta\mathbf{e}_{t+1}^i + \mathbf{g}_{t+1}^i - \mathbf{h}_{t+1}^i\right\|^2\right] \\
&= \frac{1}{n}\sum_{i=1}^{n}\mathbb{E}\left[\left\|\eta\mathbf{e}_t^i + \eta\mathbf{g}_t^i - \eta\mathbf{h}_t^i - \eta\Delta_t^i - \mathbf{h}_t^i - \Delta_t^i + \mathbf{g}_{t+1}^i\right\|^2\right] \\
&\overset{(i)}{\leq} (1+\beta_1)\frac{1}{n}\sum_{i=1}^{n}\mathbb{E}\left[\left\|\eta\mathbf{e}_t^i + (1+\eta)\mathbf{g}_t^i - (1+\eta)\mathbf{h}_t^i - (1+\eta)\Delta_t^i\right\|^2\right] \\
&\quad + (1+\beta_1^{-1})\frac{1}{n}\sum_{i=1}^{n}\mathbb{E}\left[\left\|\mathbf{g}_{t+1}^i - \mathbf{g}_t^i\right\|^2\right] \\
&\overset{(ii)}{\leq} (1+\beta_1)\frac{1}{n}\sum_{i=1}^{n}\mathbb{E}\left[\left\|\eta\mathbf{e}_t^i + (1+\eta)\mathbf{g}_t^i - (1+\eta)\mathbf{h}_t^i - (1+\eta)\Delta_t^i\right\|^2\right] \\
&\quad + 2(1+\beta_1^{-1})\frac{1}{n}\sum_{i=1}^{n}\mathbb{E}\left[\left\|\nabla f_i(\mathbf{x}_{t+1}) - \nabla f_i(\mathbf{x}_t)\right\|^2\right] + 4(1+\beta_1^{-1})\sigma^2 \\
&\overset{(iii)}{\leq} (1+\beta_1)\frac{1}{n}\sum_{i=1}^{n}\mathbb{E}\left[\left\|\eta\mathbf{e}_t^i + (1+\eta)\mathbf{g}_t^i - (1+\eta)\mathbf{h}_t^i - (1+\eta)\Delta_t^i\right\|^2\right] \\
&\quad + 2(1+\beta_1^{-1})\widetilde{L}^2\mathbb{E}\left[\left\|\mathbf{x}_{t+1} - \mathbf{x}_t\right\|^2\right] + 4(1+\beta_1^{-1})\sigma^2, \tag{22}
\end{aligned}
$$

where in $(i)$ we use Young's inequality, in $(ii)$ we use assumption 5, and in $(iii)$ we use smoothness of $f_i$. Now we consider the first term in the above:

$$
\begin{aligned}
&\frac{1}{n}\sum_{i=1}^{n}\mathbb{E}\left[\left\|\eta\mathbf{e}_t^i + (1+\eta)\mathbf{g}_t^i - (1+\eta)\mathbf{h}_t^i - (1+\eta)\Delta_t^i\right\|^2\right] \\
&\overset{(i)}{\leq} (1+\beta_2)(1+\eta)^2\frac{1}{n}\sum_{i=1}^{n}\mathbb{E}\left[\left\|\eta\mathbf{e}_t^i + \mathbf{g}_t^i - \mathbf{h}_t^i - \Delta_t^i\right\|^2\right] + (1+\beta_2^{-1})\eta^4 E_t \\
&\overset{(ii)}{\leq} (1+\beta_2)(1+\eta)^2(1-\delta)H_t + (1+\beta_2^{-1})\eta^4 E_t,
\end{aligned}
$$

where in $(i)$ we use Young's inequality, and in $(ii)$ we use the definition of the compressor. Putting it back in (22), we have:

$$
\begin{aligned}
H_{t+1} &\leq (1+\beta_1)(1+\beta_2)(1+\eta)^2(1-\delta)H_t + (1+\beta_1)(1+\beta_2^{-1})\eta^4 E_t \\
&\quad + 2(1+\beta_1^{-1})\widetilde{L}^2\mathbb{E}\left[\left\|\mathbf{x}_{t+1} - \mathbf{x}_t\right\|^2\right] + 4(1+\beta_1^{-1})\sigma^2
\end{aligned}
$$

By Lemma 7, we can substitute $\mathbb{E}\left[\left\|\mathbf{x}_{t+1} - \mathbf{x}_t\right\|^2\right]$ and get:

$$
\begin{aligned}
H_{t+1} &\leq (1+\beta_1)(1+\beta_2)(1+\eta)^2(1-\delta)H_t + (1+\beta_1)(1+\beta_2^{-1})\eta^4 E_t \\
&\quad + 2(1+\beta_1^{-1})\widetilde{L}^2\gamma^2\left(2(1-\delta)H_t + 4\eta^2 E_t + 4LF_t + \frac{2\sigma^2}{n}\right) \\
&\quad + 4(1+\beta_1^{-1})\sigma^2 \\
&= \left[(1+\beta_1)(1+\beta_2)(1+\eta)^2(1-\delta) + 4(1+\beta_1^{-1})\widetilde{L}^2\gamma^2\right]H_t \\
&\quad + \left[(1+\beta_1)(1+\beta_2^{-1})\eta^4 + 8(1+\beta_1^{-1})\widetilde{L}^2\gamma^2\eta^2\right]E_t \\
&\quad + 8(1+\beta_1^{-1})\widetilde{L}^2L\gamma^2 F_t + \left[4(1+\beta_1^{-1}) + \frac{4(1+\beta_1^{-1})\widetilde{L}^2\gamma^2}{n}\right]\sigma^2.
\end{aligned}
$$

Since $\eta = \frac{\delta}{4k}$ for some $k \geq 1$, we must have:

$$
(1+\eta)^2(1-\delta) \leq 1 - \frac{\delta}{4}.
$$

Let us choose $\beta_1 := \frac{\delta}{16-2\delta}$ and $\beta_2 := \frac{\delta}{8-2\delta}$, so we get:

$$H_{t+1} \leq \left(1 - \frac{\delta}{16} + \frac{64\widetilde{L}^2\gamma^2}{\delta}\right)H_t + \left(\frac{8\eta^4}{\delta} + \frac{128\widetilde{L}^2\gamma^2\eta^2}{\delta}\right)E_t$$

$$+ \frac{128\widetilde{L}^2 L\gamma^2}{\delta}F_t + \frac{64}{\delta}\left(1 + \frac{\widetilde{L}^2\gamma^2}{n}\right)\sigma^2$$

$$= \left(1 - \frac{\delta}{16} + \frac{64\widetilde{L}^2\gamma^2}{\delta}\right)H_t + \left(\frac{8\delta^3}{k^4 4^4} + \frac{128\widetilde{L}^2\gamma^2\delta}{k^2 4^2}\right)E_t$$

$$+ \frac{128\widetilde{L}^2 L\gamma^2}{\delta}F_t + \frac{64}{\delta}\left(1 + \frac{\widetilde{L}^2\gamma^2}{n}\right)\sigma^2.$$

If $\gamma \leq \frac{\delta}{32\sqrt{2}\widetilde{L}}$, then:

$$H_{t+1} \leq \left(1 - \frac{\delta}{32}\right)H_t + \left(\frac{8\delta^3}{k^4 4^4} + \frac{128\widetilde{L}^2\gamma^2\delta}{k^2 4^2}\right)E_t$$

$$+ \frac{128\widetilde{L}^2 L\gamma^2}{\delta}F_t + \frac{64}{\delta}\left(1 + \frac{\widetilde{L}^2\gamma^2}{n}\right)\sigma^2$$

$\square$

Finally, we consider the Lyapunov function $\Psi := X_t + aH_t + bE_t$ where constants $a$ and $b$ are set as follows: $b := \frac{48kL\gamma^3}{\delta}$, $a := \frac{512kb}{\delta^2}$, where $k$ will be set later.

**Lemma 10.** *Let $f$ be $L$-smooth and $\mu$-strongly quasi-convex around $\mathbf{x}^\star$, and each $f_i$ be $L_i$-smooth. Let $\eta = \frac{\delta}{400}$, and $\gamma \leq \frac{\delta}{3200\sqrt{2}\widetilde{L}}$, then we have*

$$\Psi_{t+1} \leq \left(1 - \min\left\{\frac{\gamma\mu}{2}, \frac{\delta}{8850}\right\}\right)\Psi_t - \frac{\gamma}{4}F_t + \gamma^2\frac{\sigma^2}{n} + \gamma^3\frac{10^{11}L\sigma^2}{\delta^4}. \tag{23}$$

*Proof.* With $\eta = \frac{\delta}{k(4-2\delta)}$ we set $\alpha = \frac{\delta}{8k-2\delta}$ in Lemma 8, and get

$$E_{t+1} \leq \left(1 - \frac{\delta}{8k}\right)E_t + \frac{8k}{\delta}H_t$$

We can put the link between the real and virtual iterates (11) and the descent lemma 1 for $X_t$ together and get

$$\begin{aligned}
\Psi_{t+1} \leq \ & \left(1 - \frac{\gamma\mu}{2}\right)X_t - \frac{\gamma}{2}F_t + \frac{\gamma^2}{n}\sigma^2 + 3L\gamma^3 E_t \\
& + a\left[\left(1 - \frac{\delta}{32}\right)H_t + \left(\frac{8\delta^3}{k^4 4^4} + \frac{128\widetilde{L}^2\gamma^2\delta}{k^2 4^2}\right)E_t + \frac{128\widetilde{L}^2 L\gamma^2}{\delta}F_t \right. \\
& \left. + \frac{64}{\delta}\left(1 + \frac{\widetilde{L}^2\gamma^2}{n}\right)\sigma^2\right] + b\left[\left(1 - \frac{\delta}{8k}\right)E_t + \frac{8k}{\delta}H_t\right].
\end{aligned}$$

Rearranging the terms we continue as follows

$$\begin{aligned}
\Psi_{t+1} \leq \ & \left(1 - \frac{\gamma\mu}{2}\right)X_t + \left(1 - \frac{\delta}{32} + \frac{8kb}{\delta a}\right)aH_t \\
& + \left(1 - \frac{\delta}{8k} + \frac{3L\gamma^3}{b} + \frac{a}{b}\left(\frac{8\delta^3}{k^4 4^4} + \frac{128\widetilde{L}^2\gamma^2\delta}{k^2 4^2}\right)\right)bE_t \\
& - \frac{\gamma}{2}F_t + \frac{128\widetilde{L}^2 L\gamma^2 a}{\delta}F_t + \gamma^2\frac{\sigma^2}{n} + \frac{64a}{\delta}\left(1 + \frac{\widetilde{L}^2\gamma^2}{n}\right)\sigma^2.
\end{aligned}$$

Now we set $\gamma \leq \frac{\delta}{32\sqrt{2k}\widetilde{L}}$, and then we get:

$$\Psi_{t+1} \leq \left(1 - \frac{\gamma\mu}{2}\right) X_t + \left(1 - \frac{\delta}{32} + \frac{8kb}{\delta a}\right) aH_t + \left(1 - \frac{\delta}{8k} + \frac{3L\gamma^3}{b} + \frac{\delta^3 a}{16k^4 b}\right) bE_t$$

$$- \frac{\gamma}{2}F_t + \frac{128\widetilde{L}^2 L\gamma^2 a}{\delta}F_t + \gamma^2\frac{\sigma^2}{n} + \frac{64a}{\delta}\left(1 + \frac{\widetilde{L}^2\gamma^2}{n}\right)\sigma^2.$$

Let $b = \frac{48kL\gamma^3}{\delta}$ and $a = \frac{512kb}{\delta^2}$, then for the coefficient next to $H_t$ have

$$1 - \frac{\delta}{32} + \frac{8kb}{\delta a} \leq 1 - \frac{\delta}{64}.$$

For the coefficient next to $E_t$ term we have

$$1 - \frac{\delta}{8k} + \frac{3L\gamma^3}{b} + \frac{\delta^3 a}{16k^4 b} \leq 1 - \frac{\delta}{16k} + \frac{32\delta}{k^3}.$$

For the coefficient next to $F_t$ term we have

$$-\frac{\gamma}{2} + \frac{128\widetilde{L}^2 L\gamma^2 a}{\delta} \leq -\frac{\gamma}{2} + \frac{3\gamma}{4k^2}.$$

Finally, for the coefficient next to $\sigma^2$ term we have

$$\frac{64a}{\delta}\left(1 + \frac{\widetilde{L}^2\gamma^2}{n}\right) \leq \gamma^3\frac{10^7 k^2 L}{\delta^4}.$$

Therefore, combining all together and setting $k = 100$, we get:

$$\Psi_{t+1} \leq \left(1 - \frac{\gamma\mu}{2}\right) X_t + \left(1 - \frac{\delta}{64}\right) aH_t + \left(1 - \frac{\delta}{8850}\right) bE_t$$

$$- \frac{\gamma}{4}F_t + \gamma^2\frac{\sigma^2}{n} + \gamma^3\frac{10^{11} L\sigma^2}{\delta^4}$$

$$\leq \left(1 - \min\left\{\frac{\gamma\mu}{2}, \frac{\delta}{8850}\right\}\right)\Psi_t - \frac{\gamma}{4}F_t + \gamma^2\frac{\sigma^2}{n} + \gamma^3\frac{10^{11} L\sigma^2}{\delta^4}. \qquad \square$$

Now we give the precise statement of Theorem 2

**Theorem 5.** *Let $f$ be $\mu$-strongly quasi-convex around $\mathbf{x}^\star$ and $L$-smooth. Let each $f_i$ be $L_i$-smooth. Then for $\eta = \frac{\delta}{400}$, there exists a $\gamma \leq \frac{\delta}{3200\sqrt{2}\widetilde{L}}$ such that after at most*

$$T = \widetilde{\mathcal{O}}\left(\frac{\sigma^2}{\mu n\varepsilon} + \frac{\sqrt{L}\sigma}{\mu\delta^2\varepsilon^{1/2}} + \frac{\widetilde{L}}{\mu\delta}\right)$$

*iterations of Algorithm 2 it holds $\mathbb{E}\left[f(\mathbf{x}_{out}) - f^\star\right] \leq \varepsilon$, where $\mathbf{x}_{out}$ is chosen randomly from $\mathbf{x}_t \in \{\mathbf{x}_0, \ldots, \mathbf{x}_T\}$ with probabilities proportional to $(1 - \frac{\gamma\mu}{2})^{-(t+1)}$.*

*Proof.* The claim of theorem 5 follows from Lemma 10; Lemma 3 and remark 4 from (Stich, 2020). Note that by the initialization, we have $\Psi_0 = \|\mathbf{x}_0 - \mathbf{x}^\star\|^2$. Also note that by the choice of parameters $\frac{\gamma\mu}{2} \leq \frac{\delta}{8850}$. $\qquad \square$

## B.2 CONVEX SETTING

We switch to the convex regime. The considered setting differs from the previous one by setting $\mu = 0$. Then the claim of Lemma 10 changes as follows.

**Lemma 11.** *Let $f$ be $L$-smooth and quasi convex, and each $f_i$ be $L_i$-smooth. Let $\eta = \frac{\delta}{400}$, and $\gamma \leq \frac{\delta}{3200\sqrt{2}\widetilde{L}}$, then we have*

$$\Psi_{t+1} \leq \Psi_t - \frac{\gamma}{4}F_t + \gamma^2\frac{\sigma^2}{n} + \gamma^3\frac{10^{11} L\sigma^2}{\delta^4}, \tag{24}$$

*Proof.* The proof immediately follows from lemma 10 by plugging in $\mu = 0$. □

Now we give the precise statement of Theorem 3

**Theorem 6.** *Let $f$ be quasi-convex around $\mathbf{x}^\star$ and $L$-smooth. Let each $f_i$ be $L_i$-smooth. Then for $\eta = \frac{\delta}{400}$, there exists a $\gamma \leq \frac{\delta}{3200\sqrt{2}\widetilde{L}}$ such that after at most*

$$T = \mathcal{O}\left(\frac{R_0\sigma^2}{n\varepsilon^2} + \frac{\sqrt{L}R_0\sigma}{\delta^2\varepsilon^{3/2}} + \frac{\widetilde{L}R_0}{\delta\varepsilon}\right)$$

*iterations of Algorithm 2 it holds $\mathbb{E}\left[f(\mathbf{x}_{out}) - f^\star\right] \leq \varepsilon$, where $R_0 := \|\mathbf{x}_0 - \mathbf{x}^\star\|^2$ and $\mathbf{x}_{out}$ is chosen uniformly at random from $\mathbf{x}_t \in \{\mathbf{x}_0, \dots, \mathbf{x}_T\}$.*

*Proof.* The claim of theorem 6 follows from lemma 11; lemma 5 and remark 6 from (Stich & Karimireddy, 2020). Note that by the initialization $\Psi_0 = R_0$. □

### B.3 Nonconvex Setting

Now we give the small modification of lemma 7 that is used in the nonconvex setting.

**Lemma 12.** *Let $f$ be $L$-smooth, then:*

$$\mathbb{E}\left[\|\mathbf{x}_{t+1} - \mathbf{x}_t\|\right] \leq \gamma^2 \left(2(1-\delta)H_t + 4\eta^2 E_t + 4\mathbb{E}\left[\|\nabla f(\mathbf{x}_t)\|^2\right] + \frac{2\sigma^2}{n}\right). \tag{25}$$

*Proof.*

$$\begin{aligned}
\frac{1}{\gamma^2}\mathbb{E}\left[\|\mathbf{x}_{t+1} - \mathbf{x}_t\|^2\right] &= \mathbb{E}\left[\|\mathbf{h}_t + \Delta_t\|^2\right] \\
&= \mathbb{E}\left[\|\Delta_t - \eta\mathbf{e}_t - \mathbf{g}_t + \mathbf{h}_t + \eta\mathbf{e}_t + \mathbf{g}_t\|^2\right] \\
&\overset{(i)}{\leq} 2\mathbb{E}\left[\|\Delta_t - \eta\mathbf{e}_t - \mathbf{g}_t + \mathbf{h}_t\|^2\right] + 2\mathbb{E}\left[\|\eta\mathbf{e}_t + \mathbf{g}_t\|^2\right] \\
&\overset{(ii)}{\leq} 2\mathbb{E}\left[\|\Delta_t - \eta\mathbf{e}_t - \mathbf{g}_t + \mathbf{h}_t\|^2\right] + 2\mathbb{E}\left[\|\eta\mathbf{e}_t + \nabla f(\mathbf{x}_t)\|^2\right] + \frac{2\sigma^2}{n} \\
&\overset{(iii)}{\leq} \frac{2}{n}\sum_{i=1}^n \mathbb{E}\left[\|\Delta_t^i - \eta\mathbf{e}_t^i - \mathbf{g}_t^i + \mathbf{h}_t^i\|^2\right] + \frac{4\eta^2}{n}\sum_{i=1}^n \mathbb{E}\left[\|\mathbf{e}_t^i\|^2\right] \\
&\quad + 4\mathbb{E}\left[\|\nabla f(\mathbf{x}_t)\|^2\right] + \frac{2\sigma^2}{n} \\
&\overset{(iv)}{\leq} \frac{2(1-\delta)}{n}\sum_{i=1}^n \mathbb{E}\left[\|\eta\mathbf{e}_t^i + \mathbf{g}_t^i - \mathbf{h}_t^i\|^2\right] + \frac{4\eta^2}{n}\sum_{i=1}^n \mathbb{E}\left[\|\mathbf{e}_t^i\|^2\right] \\
&\quad + 4\mathbb{E}\left[\|\nabla f(\mathbf{x}_t)\|^2\right] + \frac{2\sigma^2}{n}.
\end{aligned}$$

where in $(i) - (iii)$ we use Young's inequality; in $(ii)$ we use Assumption 5, and in $(iv)$ we use the definition of the compressor. □

Next, we give a simple modification of lemma 9 in nonconvex setting as well.

**Lemma 13.** *Let $f$ be $L$-smooth, and each $f_i$ be $L_i$-smooth. Let $\eta = \frac{\delta}{4k}$ for some $k \geq 1$ and $\gamma \leq \frac{\delta}{32\sqrt{2}\widetilde{L}}$. Then we have*

$$\begin{aligned}
H_{t+1} &\leq \left(1 - \frac{\delta}{32}\right)H_t + \left(\frac{8\delta^3}{k^4 4^4} + \frac{128\widetilde{L}^2\gamma^2\delta}{k^2 4^2}\right)E_t + \frac{128\widetilde{L}^2\gamma^2}{\delta}\mathbb{E}\left[\|\nabla f(\mathbf{x}_t)\|^2\right] \\
&\quad + \frac{64}{\delta}\left(1 + \frac{\widetilde{L}^2\gamma^2}{n}\right)\sigma^2.
\end{aligned} \tag{26}$$

*Proof.* The proof is almost exactly the same to the proof of lemma 9 where the bound of $\mathbb{E}\left[\|x_{t+1} - x_t\|^2\right]$ by lemma 7 is replaced by lemma 12. □

Now we consider the Lyapunov function $\Psi_t := \widetilde{F}_t + aH_t + bE_t$ where constants $a$ and $b$ are set as follows: $b := \frac{8L^2\gamma^3}{\delta}$, $a := \frac{512kb}{\delta^2}$, where $k$ will be set later.

**Lemma 14.** *Let $f$ be $L$-smooth, and each $f_i$ be $L_i$-smooth. Let $\eta = \frac{\delta}{400}$ and $\gamma \leq \frac{\delta}{3200\sqrt{2}\widetilde{L}}$. Then we have:*

$$\Psi_{t+1} \leq \Psi_t - \frac{\gamma}{8}\mathbb{E}\left[\|\nabla f(\mathbf{x}_t)\|^2\right] + \gamma^2\frac{L\sigma^2}{2n} + \gamma^3\frac{10^{10}L^2\sigma^2}{\delta^4}. \tag{27}$$

*Proof.* Note that lemma 8 still holds in the smooth nonconvex case. Therefore, with $\eta = \frac{\delta}{4k}$, if in (20) we set $\alpha = \frac{\delta}{8k-2\delta}$, then:

$$E_{t+1} \leq \left(1 - \frac{\delta}{8k}\right)E_t + \frac{8k}{\delta}H_t.$$

Now we put all inequalities together:

$$\begin{aligned}
\Psi_{t+1} \leq\ & \widetilde{F}_t - \frac{\gamma}{4}\mathbb{E}\left[\|\nabla f(\mathbf{x}_t)\|^2\right] + \frac{\gamma^3 L^2}{2}E_t + \frac{\gamma^2 L}{2n}\sigma^2 \\
& + a\left[\left(1 - \frac{\delta}{32}\right)H_t + \left(\frac{8\delta^3}{k^4 4^4} + \frac{128\widetilde{L}^2\gamma^2\delta}{k^2 4^2}\right)E_t + \frac{128\widetilde{L}^2\gamma^2}{\delta}\mathbb{E}\left[\|\nabla f(\mathbf{x}_t)\|^2\right]\right. \\
& \left. + \frac{64}{\delta}\left(1 + \frac{2\widetilde{L}^2\gamma^2}{n}\right)\sigma^2\right] \\
& + b\left[\left(1 - \frac{\delta}{8k}\right)E_t + \frac{8k}{\delta}H_t\right] \\
=\ & \widetilde{F}_t + \left(1 - \frac{\delta}{32} + \frac{8kb}{\delta a}\right)aH_t \\
& + \left(1 - \frac{\delta}{8k} + \frac{L^2\gamma^3}{2b} + \frac{a}{b}\left(\frac{8\delta^3}{k^4 4^4} + \frac{128\widetilde{L}^2\gamma^2\delta}{k^2 4^2}\right)\right)bE_t \\
& - \frac{\gamma}{4}\mathbb{E}\left[\|\nabla f(\mathbf{x}_t)\|^2\right] + \frac{128\widetilde{L}^2\gamma^2 a}{\delta}\mathbb{E}\left[\|\nabla f(\mathbf{x}_t)\|^2\right] \\
& + \gamma^2\frac{L\sigma^2}{2n} + \frac{64a}{\delta}\left(1 + \frac{\widetilde{L}^2\gamma^2}{n}\right)\sigma^2.
\end{aligned}$$

Similar as before, we now set $\gamma \leq \frac{\delta}{32\sqrt{2}k\widetilde{L}}$ and get:

$$\begin{aligned}
\Psi_{t+1} \leq\ & \widetilde{F}_t + \left(1 - \frac{\delta}{32} + \frac{8kb}{\delta a}\right)aH_t + \left(1 - \frac{\delta}{8k} + \frac{L^2\gamma^3}{2b} + \frac{\delta^3 a}{16k^4 b}\right)bE_t \\
& - \frac{\gamma}{4}\mathbb{E}\left[\|\nabla f(\mathbf{x}_t)\|^2\right] + \frac{128\widetilde{L}^2\gamma^2 a}{\delta}\mathbb{E}\left[\|\nabla f(\mathbf{x}_t)\|^2\right] + \gamma^2\frac{L}{2n}\sigma^2 + \frac{64a}{\delta}\left(1 + \frac{\widetilde{L}^2\gamma^2}{n}\right)\sigma^2.
\end{aligned}$$

Let $b = \frac{8kL^2\gamma^3}{\delta}$ and $a = \frac{512kb}{\delta^2}$, then for the coefficient next to $H_t$ term, we have

$$1 - \frac{\delta}{32} + \frac{8kb}{\delta a} \leq 1 - \frac{\delta}{64};$$

for the coefficient next to $E_t$ term, we have

$$1 - \frac{\delta}{8k} + \frac{L^2\gamma^3}{2b} + \frac{\delta^3 a}{16k^4 b} \leq 1 - \frac{\delta}{16k} + \frac{\delta^3 a}{16k^4 b} \leq 1 - \frac{\delta}{16k} + \frac{32\delta}{k^3};$$

for the coefficient next to $\mathbb{E}\left[\|\nabla f(\mathbf{x}_t)\|^2\right]$ term, we have

$$-\frac{\gamma}{4} + \frac{128\widetilde{L}^2\gamma^2 a}{\delta} \leq -\frac{\gamma}{4} + \frac{\gamma}{8k^2};$$

for the coefficient next to $\sigma^2$ term, we have

$$\frac{64a}{\delta}\left(1 + \frac{\widetilde{L}^2\gamma^2}{n}\right) \leq \gamma^3\frac{10^6 k^2 L^2}{\delta^4}.$$

Therefore, if we set $k = 100$, we get:

$$\Psi_{t+1} \leq \widetilde{F}_t + \left(1 - \frac{\delta}{64}\right)aH_t + \left(1 - \frac{\delta}{8850}\right)bE_t$$
$$- \frac{\gamma}{8}\mathbb{E}\left[\|\nabla f(\mathbf{x}_t)\|^2\right] + \gamma^2\frac{L\sigma^2}{2n} + \gamma^3\frac{10^{10}L^2\sigma^2}{\delta^4}$$
$$\leq \Psi_t - \frac{\gamma}{8}\mathbb{E}\left[\|\nabla f(\mathbf{x}_t)\|^2\right] + \gamma^2\frac{L\sigma^2}{2n} + \gamma^3\frac{10^{10}L^2\sigma^2}{\delta^4}. \qquad \square$$

Now we give the precise statement of Theorem 4

**Theorem 7.** *Let $f$ be $L$-smooth, and each $f_i$ be $L_i$-smooth. Then for $\eta = \frac{\delta}{400}$, there exists a $\gamma \leq \frac{\delta}{3200\sqrt{2}\widetilde{L}}$ such that after at most*

$$T = \mathcal{O}\left(\frac{LF_0\sigma^2}{n\varepsilon^2} + \frac{LF_0\sigma}{\delta^2\varepsilon^{3/2}} + \frac{\widetilde{L}F_0}{\delta\varepsilon}\right)$$

*iterations of Algorithm 2 it holds $\mathbb{E}\left[\|\nabla f(\mathbf{x}_{out})\|^2\right] \leq \varepsilon$, where $F_0 := f(\mathbf{x}_0) - f^\star$ and $\mathbf{x}_{out}$ is chosen uniformly at random from $\mathbf{x}_t \in \{\mathbf{x}_0, \ldots, \mathbf{x}_T\}$.*

*Proof.* The claim of theorem 7 follows from lemma 14; lemma 5 and remark 6 from (Stich, 2020). Note that by our choice of initialization, $\Psi_0 = F_0$. $\qquad \square$

**Remark 15.** *Notice that we do not explicitly bound the distance between $\mathbf{h}_t^i$ and the full local gradient $\nabla f_i(\mathbf{x}_t)$ like in the proof of EF21 (see also Appendix G). Instead, we only control the size of the error term and the compressed message. Interestingly, this way we also indirectly control the distance between $\mathbf{h}_t^i$ and the local **stochastic** gradient:*

$$\frac{1}{n}\sum_{i=1}^n \mathbb{E}\left[\|g_t^i - h_t^i\|^2\right] \leq 2\frac{1}{n}\sum_{i=1}^n \mathbb{E}\left[\|\eta e_t^i + g_t^i - h_t^i\|^2\right] + 2\frac{1}{n}\sum_{i=1}^n \eta^2\mathbb{E}\left[\|e_t^i\|^2\right]$$
$$= 2H_t + 2\eta^2 E_t$$

*With the choice $\eta \sim \delta$, $H_t + \eta^2 E_t$ is proportional to the $aH_t + bE_t$ term involved in the Lyapunov function we use in the analysis. Taking into account the convergence guarantees for the Lyapunov function, the above derivations show that the distance between $\mathbf{h}_t^i$ and $\mathbf{g}_t^i$ indeed is controled and does not blow up.*

## C   IMPORTANCE OF $\eta$ CHOICE

### C.1   THE ROLE OF $\eta$ FROM A THEORY PERSPECTIVE

Here we review the role of $\eta$ in the lemmas, and shine some light on why the choice $\eta \sim \delta$ is important. For simplicity, we only consider the strongly convex case.

- The descent of $E_t$ is achieved through $\eta$. In particular, in Lemma 8, $E_t$ is scaled by $(1 + \alpha)(1 - \eta)^2$. For $\eta \sim \delta$ and appropriate choice of $\alpha$, we get a descent on $E_t$ at the scale $(1 - \Omega(\delta))$, which is proportional to the descent of $H_t$ in Lemma 9.

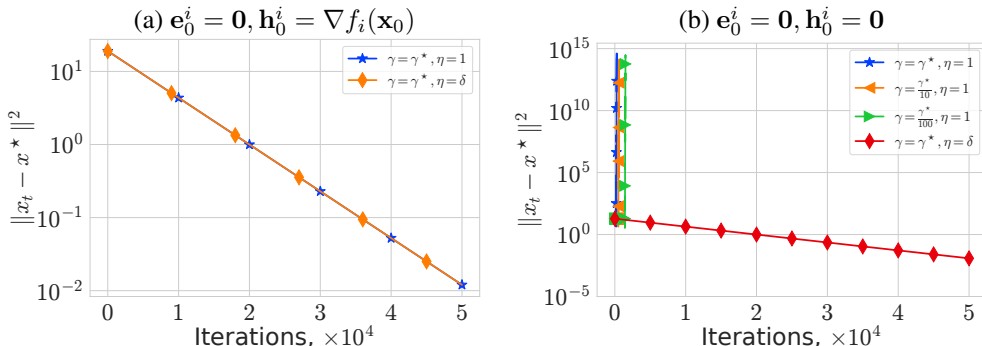

Figure 6: The convergence of EControl with $\eta = 1$ and $\eta = 0.5$ for two different initializations. EControl with $\eta = \delta$ converges regardless of the initialization while EControl with $\eta = 1$ is sensitive to the initialization. Here $\gamma^\star = \frac{\delta}{3200\sqrt{2}L}$, i.e. theoretical value of the stepsize.

- Perhaps more importantly, $\eta$ controls the contribution of $E_t$, and balances the scale between $E_t$ and $H_t$. Crucially, an $E_t$ term is introduced into the descent of $H_t$ in Lemma 9 via the upper bound on $\mathbb{E}\left[\|\mathbf{x}_t - \mathbf{x}_{t+1}\|^2\right]$ (Lemma 7). With the $\eta$ term, the contribution of $E_t$ from $\mathbb{E}\left[\|\mathbf{x}_t - \mathbf{x}_{t+1}\|^2\right]$ is of the order $\mathcal{O}(\gamma^2\eta^2/\delta)$. This turned out to be extremely important in Lemma 10, where $b \sim \delta^2 a$ (recall that $a$ is the coefficient of $H_t$ and $b$ is the coefficient of $E_t$ in the Lyapunov function $\Psi$). Taking a closer look at Lemma 10, we see that setting $\eta \sim \delta$ scales the $E_t$ term from the descent of $H_t$ by $\frac{a\gamma^2\eta^2}{b\delta} \sim \frac{\gamma^2}{\delta}$. This allows us to pick $\gamma \sim \delta$ resulting in a $\delta$ scale in front of the $E_t$ term from the descent of $H_t$.

## C.2 Unstable behavior when $\eta = 1$

Let us first consider a simple problem with $n = 2, d = 3$ where $f_1$ and $f_2$ are defined as follows

$$f_1(\mathbf{x}) = (1, 1, 5)^\top \mathbf{x} + \frac{1}{2}\|\mathbf{x}\|^2, \quad f_2(\mathbf{x}) = (1, 5, 1)^\top \mathbf{x} + \frac{1}{2}\|\mathbf{x}\|^2.$$

Obviously, this problem is strongly convex. We show that $\eta$ parameter plays an essential role in stabilizing the convergence of EControl. We show that if $\eta = 1$, then with the specific choice of initialization EControl may diverge while with $\eta = \delta$ it converges.

In the first set of experiments we consider $\mathbf{e}_0^i = \mathbf{0}, \mathbf{h}_0^i = \mathbf{0}$, while in the second one we initialize as $\mathbf{e}_0^i = \mathbf{0}, \mathbf{h}_0^i = \nabla f_i(\mathbf{x}_0)$. For simplicity, we use full gradients in both cases in order not to be affected by the noise. We apply Top-1 compression operator in all the cases, i.e. $\delta = \frac{1}{3}$. For EControl with $\eta = \delta$ we set $\gamma$ according to Theorem 2.

We demonstrate the convergence in both cases for EControl with $\eta = 1$ and $\eta = \delta$. The results are presented in Figure 6. We observe that in both cases EControl with $\eta = \delta$ converges linearly to the solution as it is predicted by our theory. In contrast, EControl with $\eta = 1$ converges only if $\mathbf{h}_0^i = \nabla f_i(\mathbf{x}_0)$ and diverges if $\mathbf{h}_0^i = \mathbf{0}$ regardless of the choice of $\gamma$. Very small values of the stepsize postpone the gradient norm blow up. This example illustrates that $\eta \sim \delta$ makes the algorithm converge more stable, i.e. it is necessary for efficient performance.

## C.3 Comparison with EF21 and diminishing $\eta$

Note that in Algorithm 2, when $\eta \to 0$, the algorithm recovers EF21. However, the crucial difference between EControl and EF21 is precisely the fact that each update is injected the scaled error term, which is the key ingredient for achieving the linear speedup in the number of clients. In this section we use the synthetic least squares problem in Section 7.1 to demonstrate that, as $\eta \to 0$, the performance of EControl degrades in the stochastic regime and its performance converges to that of EF21. The results are summarized in Figure 7.

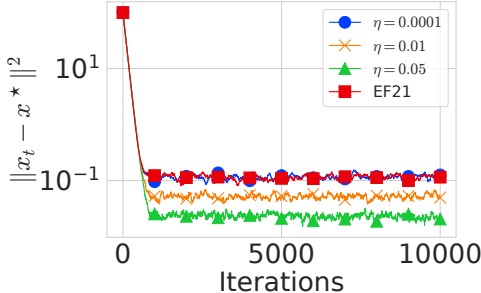

Figure 7: The convergence of ECOntrol with changing $\eta$ and EF21. $d = 200, \zeta = 100, \sigma = 50$ and number of client is 16. We apply Top-K compressor with $K/d = 0.1$. The stepsize is fixed $\gamma = 0.001$ for the purpose of illustration. EControl the performance degrades as $\eta \to 0$.

---

**Algorithm 3** EC-Approximate: EC with Approximate Bias Correction

---

1: **Input:** $\mathbf{x}_0, \gamma, \mathbf{e}_t^i = \mathbf{0}_d, \mathcal{C}_\delta$, and $\{\mathbf{h}^i\}_{i \in [n]}$ such that $\frac{1}{n} \sum_{i=1}^n \mathbb{E}\left[\left\|\mathbf{h}^i - \nabla f_i(\mathbf{x}^\star)\right\|^2\right] \leq \alpha$
2: $\mathbf{h} = \frac{1}{n} \sum_{i=1}^n \mathbf{h}^i$
3: **for** $t = 0, 1, 2, \dots$ **do**
4: $\quad \mathbf{g}_t^i = \mathbf{g}_t^i$ $\hfill \triangledown$ client side
5: $\quad \hat{\Delta}_t^i = \mathcal{C}_\delta(\mathbf{e}_t^i + \mathbf{g}_t^i - \mathbf{h}^i)$
6: $\quad \mathbf{e}_{t+1}^i = \mathbf{e}_t^i + \mathbf{g}_t^i - \mathbf{h}^i - \hat{\Delta}_t^i$
7: $\quad$ send to server: $\hat{\Delta}_t^i$
8: $\quad \mathbf{x}_{t+1} := \mathbf{x}_t - \gamma \mathbf{h} - \frac{\gamma}{n} \sum_{i=1}^n \hat{\Delta}_t^i$ $\hfill \triangledown$ server side

---

## D  EC-Approximate: APPROXIMATE BIAS CORRECTION

In Section 4 we described the ideal version of EC when we have access to $\mathbf{h}_\star^i = \nabla f_i(\mathbf{x}^\star)$. However, in practice, it is not known most of the time. Therefore, we propose to use a good enough approximation instead to make the method implementable. This idea leads to EC-Approximate summarised in Algorithm 3. There, instead of $\mathbf{h}_\star^i$ we use its estimator $\mathbf{h}^i$ which should estimate the true value $\mathbf{h}_\star^i$ well enough. In more details, we run EC with $\mathbf{h}^i$ satisfying

$$\frac{1}{n} \sum_{i=1}^n \mathbb{E}\left[\left\|\mathbf{h}^i - \nabla f_i(\mathbf{x}^\star)\right\|^2\right] \leq \alpha,$$

in other words, the average distance between $\mathbf{h}^i$ and $\nabla f_i(\mathbf{x}^\star)$ is at most $\alpha$. A trivial choice of $\mathbf{h}$ is the zero vectors, for which Algorithm 3 recovers the D-EC-SGD (Stich, 2020) algorithm with bounded gradient (at optimum) assumption. However, the smaller $\alpha$ is (i.e., the better approximation we have), the better convergence is. Thus, it is beneficial to obtain $\mathbf{h}^i$ after a preprocessing. For example, a nontrivial choice would be some $\mathbf{h}^i$ such that $\frac{1}{n} \sum_{i=1}^n \mathbb{E}\left[\left\|\mathbf{h}^i - \nabla f_i(\mathbf{x}^\star)\right\|^2\right] \leq \mathcal{O}(\frac{\sigma^2 L}{\delta^2 \mu})$. Such $\{\mathbf{h}^i\}_{i \in [n]}$ can be obtained by running EF21 for $\widetilde{\mathcal{O}}(\frac{L}{\delta \mu})$ rounds, which is constant and does not depend on the accuracy $\varepsilon$. We show that it is indeed possible in Appendix G for completeness.

### D.1  CONVERGENCE ANALYSIS

In this section we prove the convergence of Algorithm 3. The next lemma bounds the descent of $E_t$.

**Lemma 16.** *Let $f_i$ be $L$-smooth and convex. Let $\{\mathbf{h}^i\}_{i \in [n]}$ be such that $\frac{1}{n} \sum_{i=1}^n \mathbb{E}\left[\left\|\mathbf{h}^i - \nabla f_i(\mathbf{x}^\star)\right\|^2\right] \leq \alpha$, then the iterates of EC-Approximate satisfy*

$$E_{t+1} \leq \left(1 - \frac{\delta}{2}\right) E_t + \frac{8L}{\delta} F_t + \frac{4\alpha}{\delta} + \sigma^2. \tag{28}$$

*Proof.*

$$E_{t+1} = \frac{1}{n} \sum_{i=1}^{n} \mathbb{E}\left[\left\|\mathbf{e}_t^i + \mathbf{g}_t^i - \mathbf{h}^i - C_\delta(\mathbf{e}_t^i + \mathbf{g}_t^i - \mathbf{h}^i)\right\|^2\right]$$

$$\leq \frac{(1-\delta)}{n} \sum_{i=1}^{n} \mathbb{E}\left[\left\|\mathbf{e}_t^i + \mathbf{g}_t^i - \mathbf{h}^i\right\|^2\right]$$

$$= \frac{(1-\delta)}{n} \sum_{i=1}^{n} \mathbb{E}\left[\left\|\mathbf{e}_t^i + \nabla f_i(\mathbf{x}_t) - \mathbf{h}^i\right\|^2\right] + \sigma^2$$

$$\leq \frac{(1-\delta)(1+\beta)}{n} \sum_{i=1}^{n} \mathbb{E}\left[\left\|\mathbf{e}_t^i\right\|^2\right]$$

$$+ \frac{(1-\delta)(1+\beta^{-1})}{n} \sum_{i=1}^{n} \mathbb{E}\left[\left\|\nabla f_i(\mathbf{x}_t) - \mathbf{h}^i\right\|^2\right] + \sigma^2.$$

where in the last inequality we used Young's inequality. Setting $\beta = \frac{\delta}{2(1-\delta)}$, we have:

$$E_{t+1} \leq \left(1 - \frac{\delta}{2}\right) E_t + \frac{4}{\delta} \frac{1}{n} \sum_{i=1}^{n} \mathbb{E}\left[\left\|\nabla f_i(\mathbf{x}_t) - \nabla f_i(\mathbf{x}^\star)\right\|^2\right]$$

$$+ \frac{4}{\delta} \frac{1}{n} \sum_{i=1}^{n} \mathbb{E}\left[\left\|\nabla f_i(\mathbf{x}^\star) - \mathbf{h}^i\right\|^2\right] + \sigma^2$$

$$\leq \left(1 - \frac{\delta}{2}\right) E_t + \frac{8L}{\delta} F_t + \frac{4\alpha}{\delta} + \sigma^2.$$

where for the first inequality we used Young's inequality again, and for the second inequality we used the smoothness and convexity of $f_i$, and our assumption on $\mathbf{h}^i$. $\qquad\square$

Now consider the Lyapunov function $\Psi_t = X_t + aE_t$ for $a := \frac{12L\gamma^3}{\delta}$. We obtain the following descent lemma in $\Psi_t$.

**Lemma 17.** *Let $f$ be $\mu$-strongly quasi-convex around $\mathbf{x}^\star$, and each $f_i$ be $L$-smooth and convex. Let $\{\mathbf{h}^i\}_{i \in [n]}$ be such that $\frac{1}{n} \sum_{i=1}^{n} \mathbb{E}\left[\left\|\mathbf{h}^i - \nabla f_i(\mathbf{x}^\star)\right\|^2\right] \leq \alpha$, and the stepsize be $\gamma \leq \frac{\delta}{8\sqrt{6}L}$. Then the iterates of* EC-Approximate *satisfy*

$$\Psi_{t+1} \leq (1-c)\Psi_t - \frac{\gamma}{4}F_t + \gamma^2 \frac{\sigma^2}{n} + \gamma^3 \left(\frac{48L\alpha}{\delta^2} + \frac{12L\sigma^2}{\delta}\right) \qquad (29)$$

*where $a := \frac{12L\gamma^3}{\delta}$ and $c := \frac{\gamma\mu}{2}$*

*Proof.* Putting Lemma 1 and Lemma 16 together, we have:

$$\Psi_{t+1} \leq \left(1 - \frac{\gamma\mu}{2}\right) X_t - \frac{\gamma}{2} F_t + \gamma^2 \frac{\sigma^2}{n} + 3L\gamma^3 E_t$$

$$+ a\left(\left(1 - \frac{\delta}{2}\right) E_t + \frac{8L}{\delta} F_t + \frac{4\alpha}{\delta} + \sigma^2\right)$$

$$= \left(1 - \frac{\gamma\mu}{2}\right) X_t + \left(1 - \frac{\delta}{2} + \frac{3L\gamma^3}{a}\right) aE_t$$

$$+ \gamma^2 \frac{\sigma^2}{n} + a\left(\frac{4\alpha}{\delta} + \sigma^2\right)$$

$$- \left(\frac{\gamma}{2} - \frac{8La}{\delta}\right) F_t.$$

Plugging in the choice of $a$, we have for the $E_t$ term:

$$1 - \frac{\delta}{2} + \frac{3L\gamma^3}{a} = 1 - \frac{\delta}{4},$$

and for $F_t$ term:

$$\frac{\gamma}{2} - \frac{8La}{\delta} = \frac{\gamma}{2} - \gamma\frac{96L^2\gamma^2}{\delta^2} \geq \frac{\gamma}{2}.$$

Combining the above together we derive the statement of the lemma. ☐

**Theorem 8** (EC-Approximate: EC with approximate bias correction). *Let $f : \mathbb{R}^d \to \mathbb{R}$ be $\mu$-strongly quasi-convex, and each $f_i$ be $L$-smooth and convex. Let $\{\mathbf{h}^i\}_{i\in[n]}$ be such that $\frac{1}{n}\sum_{i=1}^n \mathbb{E}\left[\left\|\mathbf{h}^i - \nabla f_i(\mathbf{x}^\star)\right\|^2\right] \leq \alpha$. Let stepsize be $\gamma \leq \frac{\delta}{8\sqrt{6}L}$. Then after at most*

$$T = \widetilde{\mathcal{O}}\left(\frac{\sigma^2}{\mu n\varepsilon} + \frac{\sqrt{L}\sigma}{\mu\sqrt{\delta}\sqrt{\varepsilon}} + \frac{\sqrt{L\alpha}}{\mu\delta\sqrt{\varepsilon}} + \frac{L}{\mu\delta}\right)$$

*iterations of Algorithm 3 it holds $\mathbb{E}\left[f(\mathbf{x}_{out}) - f^\star\right] \leq \varepsilon$, where $\mathbf{x}_{out}$ is chosen randomly from $\{\mathbf{x}_0, \ldots, \mathbf{x}_T\}$ with probabilities proportional to $(1 - \mu\gamma/2)^{-(t+1)}$.*

*Proof.* We need to apply the results of Lemma 17, Lemma 3, and Remark 4 noticing that $\frac{\gamma\mu}{2} \leq \frac{\delta}{2}$ always due to the choice of the stepsize. ☐

According to the statement of Theorem 8, the inaccuracy in the approximation affects the higher order terms only. We clearly see that EC-Approximate achieves optimal sample complexity. This result suggests that preprocessing in the beginning of the training for a constant number of iterations may lead to better convergence guarantees in comparison with original EC.

In Appendix G we show that in a constant number of rounds of preprocessing using EF21, one can obtain a good $\{\mathbf{h}^i\}_{i\in[n]}$ with error of the order $\mathcal{O}(\sigma^2)$ for D-EC-SGD with approximate bias correction. We summarize it in the following corollary.

**Corollary 18.** *Let $f : \mathbb{R}^d \to \mathbb{R}$ be $\mu$-strongly quasi-convex, and each $f_i$ be $L$-smooth and convex. Let run EF21 as a preprocessing for $\mathcal{O}\left(\frac{L}{\delta\mu}\log\frac{LF_0\delta}{\sigma^2\mu}\right)$ rounds. Let the stepsize be $\gamma \leq \frac{\delta}{8\sqrt{6}L}$. Then after at most*

$$T = \widetilde{\mathcal{O}}\left(\frac{\sigma^2}{\mu n\varepsilon} + \frac{L\sigma}{\mu^{3/2}\delta^2\sqrt{\varepsilon}} + \frac{L}{\mu\delta}\right)$$

*iterations of Algorithm 3 it holds $\mathbb{E}\left[f(\mathbf{x}_{out}) - f^\star\right] \leq \varepsilon$, where $\mathbf{x}_{out}$ is chosen randomly from $\{\mathbf{x}_0, \ldots, \mathbf{x}_T\}$, chosen with probabilities proportional to $(1 - \mu\gamma/2)^{-(t+1)}$.*

*Proof.* We only need to apply the results of Lemma 25 and Lemma 26 in Theorem 8. ☐

## E  CONVERGENCE OF EC-Ideal

In this section we derive the convergence of EC-Ideal. The result directly follows from Theorem 8 with $\alpha = 0$.

**Theorem 1** (Convergence of EC-Ideal). *Let $f : \mathbb{R}^d \to \mathbb{R}$ be $\mu$-strongly quasi-convex around $\mathbf{x}^\star$, and each $f_i$ be $L$-smooth and convex. Then there exists a stepsize $\gamma \leq \frac{\delta}{8\sqrt{6}L}$ such that after at most*

$$T = \widetilde{\mathcal{O}}\left(\frac{\sigma^2}{\mu n\varepsilon} + \frac{\sqrt{L}\sigma}{\mu\sqrt{\delta}\varepsilon^{1/2}} + \frac{L}{\mu\delta}\right)$$

*iterations of Algorithm 1 it holds $\mathbb{E}\left[f(\mathbf{x}_{out}) - f^\star\right] \leq \varepsilon$, where $\mathbf{x}_{out}$ is chosen randomly from $\{\mathbf{x}_0, \ldots, \mathbf{x}_T\}$ with probabilities proportional to $(1 - \mu\gamma/2)^{-(t+1)}$.*

*Proof.* We need to apply the results of Theorem 8 with $\alpha = 0$. ☐

---

**Algorithm 4** D-EC-SGD with Bias Correction and Double Contractive Compression

---

1: **Input:** $\mathbf{x}_0, \mathbf{e}_0^i = \mathbf{0}_d, \mathbf{h}_0^i = \mathbf{g}_0^i, \mathbf{h}_0 = \frac{1}{n}\sum_{i=1}^n \mathbf{h}_0^i, \mathcal{C}_{\delta_1}, \mathcal{C}_{\delta_2}$, and $\gamma$
2: **for** $t = 0, 1, 2, \ldots$ **do**
3: $\quad\big|\quad$ compute $\mathbf{g}_t^i = \mathbf{g}_t^i$ $\hfill \triangledown$ client side
4: $\quad\big|\quad$ compute $\Delta_t^i = \mathcal{C}_{\delta_1}(\mathbf{e}_t^i + \mathbf{g}_t^i - \mathbf{h}_t^i)$ and $\hat{\Delta}_t^i = \mathcal{C}_{\delta_2}(\mathbf{g}_t^i - \mathbf{h}_t^i)$
5: $\quad\big|\quad$ update $\mathbf{e}_{t+1}^i = \mathbf{e}_t^i + \mathbf{g}_t^i - \mathbf{h}_t^i - \Delta_t^i$ and $\mathbf{h}_{t+1}^i = \mathbf{h}_t^i + \hat{\Delta}_t^i$
6: $\quad\big|\quad$ send to server $\Delta_t^i$ and $\hat{\Delta}_t^i$
7: $\quad\big|\quad$ update $\mathbf{x}_{t+1} := \mathbf{x}_t - \gamma\mathbf{h}_t - \frac{\gamma}{n}\sum_{i=1}^n \Delta_t^i$ $\hfill \triangledown$ server side
8: $\quad\big|\quad$ update $\mathbf{h}_{t+1} = \mathbf{h}_t + \frac{1}{n}\sum_{i=1}^n \hat{\Delta}_t^i$

---

# F   D-EC-SGD WITH BIAS CORRECTION AND DOUBLE CONTRACITVE COMPRESSION

In this section we follow algorithm D-EC-SGD with bias correction (Stich, 2020). In this algorithm, the learning mechanism for $\mathbf{h}_t^i$ to approximate $\mathbf{h}_\star^i$ is built using additional compressor from more restricted class of unbiased compression operators. We show that unbiased compressor can be replaced by more general contractive one; see Algorithm 4 for more detailed description.

## F.1   NOTATION

For D-EC-SGD with bias correction and double contractive compression we consider the following notation

$$X_t = \mathbb{E}\left[\|\tilde{\mathbf{x}}_t - \mathbf{x}^\star\|^2\right], \quad E_t = \frac{1}{n}\sum_{i=1}^n \mathbb{E}\left[\|\mathbf{e}_t^i\|^2\right], \quad \text{and} \quad H_t = \frac{1}{n}\sum_{i=1}^n \mathbb{E}\left[\|\nabla f(\mathbf{x}_t) - \mathbf{h}_t^i\|^2\right].$$

(30)

## F.2   CONVERGENCE ANALYSIS

Now we present the convergence rate for D-EC-SGD with bias correction and double contractive compression.

First, we highlight that Lemma 1. Next, we present the descent lemma in $E_t$.

**Lemma 19.** *The iterates of* D-EC-SGD *with bias correction and double contractive compression satisfy*

$$E_{t+1} \leq (1 - \delta/2)E_t + \frac{2(1 - \delta_1)}{\delta_1}H_t + (1 - \delta_1)\sigma^2.$$

(31)

*Proof.* We have

$$
\begin{aligned}
E_{t+1} &= \frac{1}{n}\sum_{i=1}^n \mathbb{E}\left[\|\mathbf{e}_{t+1}^i\|^2\right] \\
&= \frac{1}{n}\sum_{i=1}^n \mathbb{E}\left[\|\mathbf{e}_t^i + \mathbf{g}_t^i - \mathbf{h}_t^i - \mathcal{C}_{\delta_1}(\mathbf{e}_t^i + \mathbf{g}_t^i - \mathbf{h}_t^i)\|^2\right] \\
&\leq \frac{(1 - \delta_1)}{n}\sum_{i=1}^n \mathbb{E}\left[\|\mathbf{e}_t^i + \mathbf{g}_t^i - \mathbf{h}_t^i\|^2\right] \\
&\leq \frac{(1 - \delta_1)}{n}\sum_{i=1}^n \mathbb{E}\left[\|\mathbf{e}_t^i + \nabla f_i(\mathbf{x}_t) - \mathbf{h}_t^i\|^2\right] + (1 - \delta_1)\sigma^2.
\end{aligned}
$$

Using Young's inequality we continue

$$
\begin{aligned}
E_{t+1} &\leq (1 - \delta_1)(1 + \alpha_1)E_t + (1 - \delta_1)(1 + \alpha_1^{-1})\frac{1}{n}\sum_{i=1}^n \mathbb{E}\left[\|\nabla f_i(\mathbf{x}_t) - \mathbf{h}_t^i\|^2\right] \\
&\quad + (1 - \delta_1)\sigma^2.
\end{aligned}
$$

Now, if we choose $\alpha_1 = \frac{\delta_1}{2(1-\delta_1)}$, we obtain the statement of the lemma. $\qquad\square$

**Lemma 20.** *Let $f$ be $L$-smooth, then the iterates of* D-EC-SGD *with bias correction and double contractive compression satisfy*

$$\frac{1}{\gamma^2}\mathbb{E}\left[\|\mathbf{x}_{t+1} - \mathbf{x}_t\|^2\right] \leq 4\sigma^2 + 4H_t + 8E_t + 8LF_t. \tag{32}$$

*Proof.* We have

$$
\begin{aligned}
\frac{1}{\gamma^2}\mathbb{E}\left[\|\mathbf{x}_{t+1} - \mathbf{x}_t\|^2\right] &= \mathbb{E}\left[\|\mathbf{h}^t \pm \mathbf{e}_t \pm \mathbf{g}_t + \frac{1}{n}\sum_{i=1}^n \Delta_t^i\|^2\right] \\
&\leq 2\mathbb{E}\left[\|\mathbf{h}_t - \mathbf{e}_t - \mathbf{g}_t + \frac{1}{n}\sum_{i=1}^n \mathcal{C}_{\delta_1}(\mathbf{e}_t^i + \mathbf{g}_t^i - \mathbf{h}_t^i)\|^2\right] + 2\mathbb{E}\left[\|\mathbf{e}_t + \mathbf{g}_t\|^2\right] \\
&\leq \frac{2}{n}\sum_{i=1}^n \mathbb{E}\left[\|\mathbf{h}_t^i - \mathbf{e}_t^i - \mathbf{g}_t^i + \mathcal{C}_{\delta_1}(\mathbf{e}_t^i + \mathbf{g}_t^i - \mathbf{h}_t^i)\|^2\right] + 2\frac{\sigma^2}{n} + 4E_t \\
&\quad + 4\mathbb{E}\left[\|\nabla f(\mathbf{x}_t)\|^2\right] \\
&\leq \frac{2(1-\delta_1)}{n}\sum_{i=1}^n \mathbb{E}\left[\|\mathbf{h}_t^i - \mathbf{e}_t^i - \mathbf{g}_t^i\|^2\right] + 2\frac{\sigma^2}{n} + 4E_t + 8LF_t \\
&\leq 2(1-\delta_1)\sigma^2 + 4(1-\delta_1)E_t + 4(1-\delta_1)H_t + 2\frac{\sigma^2}{n} + 4E_t + 4LF_t \\
&\leq 4\sigma^2 + 4H_t + 8E_t + 8LF_t. \qquad\square
\end{aligned}
$$

In our next lemma we state the descent in $H_t$.

**Lemma 21.** *Let $f$ be $L$-smooth and each $f_i$ be $L_i$-smooth, then the iterates of* D-EC-SGD *with bias correction and double contractive compression satisfy*

$$H_{t+1} \leq (1 - \frac{\delta_2}{4} + \frac{16\tilde{L}^2\gamma^2}{\delta_2})H_t + \frac{32\tilde{L}^2\gamma^2}{\delta_2}E_t + \frac{16\tilde{L}^2L\gamma^2}{\delta_2}F_t + \left(\frac{16\tilde{L}^2\gamma^2}{\delta_2} + \frac{8}{\delta_2}\right)\sigma^2$$

*In particular, if $\gamma \leq \frac{\delta_2}{8\sqrt{2}\tilde{L}}$, then:*

$$H_{t+1} \leq (1 - \frac{\delta_2}{8})H_t + \frac{32\tilde{L}^2\gamma^2}{\delta_2}E_t + \frac{16\tilde{L}^2L\gamma^2}{\delta_2}F_t + \left(\frac{16\tilde{L}^2\gamma^2}{\delta_2} + \frac{8}{\delta_2}\right)\sigma^2 \tag{33}$$

*Proof.* We have taking the expectation $\mathbb{E}_t[\cdot]$ w.r.t $\mathbf{x}_t$:

$$\mathbb{E}_t[H_{t+1}] = \frac{1}{n}\sum_{i=1}^{n}\mathbb{E}_t\left[\left\|\nabla f_i(\mathbf{x}_{t+1}) - \mathbf{h}_t^i - \mathcal{C}_{\delta_2}(\mathbf{g}_t^i - \mathbf{h}_t^i)\right\|^2\ \right]$$

$$\overset{(i)}{\leq} \frac{1}{n}\sum_{i=1}^{n}(1+\beta)\mathbb{E}_t\left[\left\|\nabla f_i(\mathbf{x}_{t+1}) - \nabla f_i(\mathbf{x}_t)\right\|^2\right]$$

$$+ \frac{1}{n}\sum_{i=1}^{n}(1+\beta^{-1})\mathbb{E}_t\left[\left\|\nabla f_i(\mathbf{x}_t) - \mathbf{h}_t^i - \mathcal{C}_{\delta_2}(\mathbf{g}_t^i - \mathbf{h}_t^i)\right\|^2\right]$$

$$\overset{(ii)}{\leq} \tilde{L}^2(1+\beta)\mathbb{E}_t\left[\|\mathbf{x}_{t+1} - \mathbf{x}_t\|^2\right]$$

$$+ \frac{1}{n}\sum_{i=1}^{n}(1+\beta^{-1})\mathbb{E}_t\left[\left\|\mathbf{g}_t^i + (\nabla f_i(\mathbf{x}_t) - \mathbf{g}_t^i) - \mathbf{h}_t^i - \mathcal{C}_{\delta_2}(\mathbf{g}_t^i - \mathbf{h}_t^i)\right\|^2\right]$$

$$\overset{(iii)}{\leq} \tilde{L}^2(1+\beta)\mathbb{E}_t\left[\|\mathbf{x}_{t+1} - \mathbf{x}_t\|^2\right]$$

$$+ \frac{1}{n}\sum_{i=1}^{n}(1+\beta^{-1})(1+s_1)\mathbb{E}_t\left[\|\mathbf{g}_t^i - \mathbf{h}_t^i - \mathcal{C}_{\delta_2}(\mathbf{g}_t^i - \mathbf{h}_t^i)\|^2\right]$$

$$+ \frac{1}{n}\sum_{i=1}^{n}(1+\beta^{-1})(1+s_1^{-1})\mathbb{E}_t\left[\left\|\nabla f_i(\mathbf{x}_t) - \mathbf{g}_t^i\right\|^2\right]$$

$$\overset{(iv)}{\leq} \tilde{L}^2(1+\beta)\,\mathbb{E}\,\|\mathbf{x}_{t+1} - \mathbf{x}_t\|^2$$

$$+ \frac{1}{n}\sum_{i=1}^{n}(1+\beta^{-1})(1+s_1)(1-\delta_2)\mathbb{E}_t\left[\|\mathbf{g}_t^i - \mathbf{h}_t^i\|^2\right] + (1+\beta^{-1})(1+s_1^{-1})\sigma^2,$$

where in $(i)$ we use Young's inequality; in $(ii)$ we $L$-smoothness; $(iii)$ we use variance decomposition; in $(iv)$ we use the definition of $\mathcal{C}_{\delta_2}$ and bounded variance assumption. Note that $\mathbf{g}_t^i$'s are independent from $\nabla f_i(\mathbf{x}_t) - \mathbf{h}_t^i$, and $\mathbb{E}_t\left[\nabla f_i(\mathbf{x}_t) - \mathbf{g}_t^i\right] = 0$, we have:

$$\mathbb{E}_t[H_{t+1}] \leq \tilde{L}^2(1+\beta)\mathbb{E}_t\left[\|\mathbf{x}_{t+1} - \mathbf{x}_t\|^2\right] + \frac{1}{n}\sum_{i=1}^{n}(1+\beta^{-1})(1+s_1)(1-\delta_2)\left\|\nabla f_i(\mathbf{x}_t) - \mathbf{h}_t^i\right\|^2$$

$$+ (1+\beta^{-1})(1+s_1)(1-\delta_2)\sigma^2 + (1+\beta^{-1})(1+s_1^{-1})\sigma^2$$

We also can upper bound $\mathbb{E}\,\|\mathbf{x}_{t+1} - \mathbf{x}_t\|^2$ by Lemma 20 which leads to

$$\mathbb{E}_t[H_{t+1}] \leq (1+\beta^{-1})(1+s_1)(1-\delta_2)H_t$$
$$+ \tilde{L}^2(1+\beta)\gamma^2\left(8E_t + 4H_t + 4LF_t + 4\sigma^2\right)$$
$$+ \left((1+\beta^{-1})(1+s_1^{-1}) + (1+\beta^{-1})(1+s_1)(1-\delta_2)\right)\sigma^2$$

Now we need to properly set all constants $\beta, s_1$, to derive the lemma statement. In particular, if we choose $\beta = \frac{4-2\delta_2}{\delta_2}$ and $s_1 = \frac{\delta_2}{2(1-\delta_2)}$, then

$$H_{t+1} \leq (1 - \frac{\delta_2}{4} + \frac{16\tilde{L}^2\gamma^2}{\delta_2})H_t + \frac{32\tilde{L}^2\gamma^2}{\delta_2}E_t + \frac{16\tilde{L}^2 L\gamma^2}{\delta_2}F_t + \left(\frac{16\tilde{L}^2\gamma^2}{\delta_2} + \frac{8}{\delta_2}\right)\sigma^2 \qquad \square$$

Next we consider the Lyapunov function $\Psi_t := X_t + aH_t + bE_t$.

**Lemma 22.** *Let $f$ be $L$-smooth and $\mu$-strongly quasi-convex, and each $f_i$ be $L_i$-smooth. If $\gamma \leq \frac{\delta_1\delta_2}{64\sqrt{2}\tilde{L}}$, then:*

$$\Psi_{t+1} \leq (1 - \frac{\gamma\mu}{2})\Psi_t - \frac{\gamma}{4}F_t + \gamma^2\frac{\sigma^2}{n} + \gamma^3\frac{3100L\sigma^2}{\delta_1^2\delta_2^2} \tag{34}$$

*where $b := \frac{12L\gamma^3}{\delta_1}$ and $a := \frac{32b}{\delta_1\delta_2}$.*

*Proof.* By Lemma 1, Lemma 19, and Lemma 21, we have:

$$\Psi_{t+1} = X_{t+1} + aH_{t+1} + bE_{t+1}$$

$$\leq (1 - \frac{\gamma\mu}{2})X_t - \frac{\gamma}{2}F_t + \frac{\gamma^2}{n}\sigma^2 + 3L\gamma^3 E_t$$

$$+ a\left((1 - \frac{\delta_2}{8})H_t + \frac{32\tilde{L}^2\gamma^2}{\delta_2}E_t + \frac{16\tilde{L}^2 L\gamma^2}{\delta_2}F_t + \left(\frac{16\tilde{L}^2\gamma^2}{\delta_2} + \frac{8}{\delta_2}\right)\sigma^2\right)$$

$$+ b\left((1 - \frac{\delta_1}{2})E_t + \frac{2(1-\delta_1)}{\delta_1}H_t + (1-\delta_1)\frac{\sigma^2}{B}\right)$$

$$= (1 - \frac{\gamma\mu}{2})X_t + \left(1 - \frac{\delta_2}{8} + \frac{2(1-\delta_1)b}{\delta_1 a}\right)aH_t$$

$$+ \left(1 - \frac{\delta_1}{2} + \frac{3L\gamma^3}{b} + \frac{32\tilde{L}^2\gamma^2 a}{\delta_2 b}\right)bE_t$$

$$+ \left(\frac{\gamma^2}{n} + a(\frac{16\tilde{L}^2\gamma^2}{\delta_2} + \frac{8}{\delta_2}) + b(1-\delta_1)\right)\sigma^2$$

$$+ \left(-\frac{\gamma}{2} + \frac{16\tilde{L}^2 L\gamma^2 a}{\delta_2}\right)F_t$$

If $b := \frac{12L\gamma^3}{\delta_1}$, $a := \frac{32b}{\delta_1\delta_2}$, and if $\gamma \leq \frac{\delta_1\delta_2}{64\sqrt{2}\tilde{L}}$, then for the coefficients next to $H_t$ we have:

$$1 - \frac{\delta_2}{8} + \frac{2(1-\delta_1)b}{\delta_1 a} \leq 1 - \frac{\delta_2}{16};$$

for the coefficients next to $E_t$ we have:

$$1 - \frac{\delta_1}{2} + \frac{3L\gamma^3}{b} + \frac{32\tilde{L}^2\gamma^2 a}{\delta_2 b} \leq 1 - \frac{\delta_1}{8};$$

for the coefficients next to $\sigma^2$ we have:

$$a(\frac{16\tilde{L}^2\gamma^2}{\delta_2} + \frac{8}{\delta_2}) + b(1-\delta_1) \leq \gamma^3\frac{3100L}{\delta_1^2\delta_2^2};$$

for the coefficients next to $F_t$ we have:

$$-\frac{\gamma}{2} + \frac{16\tilde{L}^2 L\gamma^2 a}{\delta_2} \leq -\frac{\gamma}{4}.$$

Putting it together we have:

$$\Psi_{t+1} \leq (1 - \frac{\gamma\mu}{2})\Psi_t - \frac{\gamma}{4}F_t + \gamma^2\frac{\sigma^2}{n} + \gamma^3\frac{3100L\sigma^2}{\delta_1^2\delta_2^2}$$

where we note that $\frac{\gamma\mu}{2} \leq \frac{\delta_1}{8}$ and $\frac{\gamma\mu}{2} \leq \frac{\delta_2}{16}$. □

**Theorem 9.** *Let $f$ be $\mu$-strongly quasi-convex around $\mathbf{x}^\star$ and $L$-smooth. Let each $f_i$ be $L_i$-smooth. Let $\gamma \leq \frac{\delta_1\delta_2}{64\sqrt{2}\tilde{L}}$. Then after at most*

$$T = \widetilde{\mathcal{O}}\left(\frac{\sigma^2}{\mu n\varepsilon} + \frac{\sqrt{L}\sigma}{\mu\delta_1\delta_2\varepsilon^{1/2}} + \frac{\tilde{L}}{\mu\delta_1\delta_2}\right)$$

*iterations of Algorithm 4 it holds $\mathbb{E}[f(\mathbf{x}_{out}) - f^\star] \leq \varepsilon$, where $\mathbf{x}_{out}$ is chosen randomly from $\mathbf{x}_t \in \{\mathbf{x}_0, \ldots, \mathbf{x}_T\}$ with probabilities proportional to $(1 - \frac{\gamma\mu}{2})^{-(t+1)}$.*

*Proof.* We need to apply the results of Lemma 3, Lemma 22 and Remark 4. □

We observe that D-EF-SGD with double contracitve compression still achieves nearly optimal asymptotic complexity with stochastic gradients, where a $\sigma^2$ factor is hidden in the log terms. However, in the non-asymptotic regime it has poor dependency on compression parameters $\delta_1$ and $\delta_2$. In the simplest full gradient case, when $\delta_1 = \delta_2 = \delta$ and $\sigma^2 = 0$, the linearly convergent term is proportional to $\delta^{-2}$. In opposite, EF21 and EControl have only $\delta^{-1}$ dependency in this setting.

## G  COMPLEXITY OF EF21-SGD

In this section we consider EF21 mechanism. Richtárik et al. (2021) demonstrate that EF21-GD, i.e. EF21 with full local gradient computations converges linearly. In the stochastic setting, it has been shown in (Fatkhullin et al., 2023) that EF21-SGD converges only with large batches. For completeness, we present convergence guarantees of EF21 with arbitrary batch size. In particular, we show that EF21 can converge to an error of $\mathcal{O}(\frac{\sigma^2 L}{\delta^2 \mu})$ in a log number of rounds. EF21 is summarized in Algorithm 5.

---

**Algorithm 5** EF21

---

1: **Input:** $\mathbf{x}_0, \mathbf{h}_0^i = \nabla f_i(\mathbf{x}_0)$, $\gamma$, and $\mathbf{h}_t = \frac{1}{n} \sum_{i=1}^n \mathbf{h}_t^i$
2: **for** $t = 0, 1, 2, \dots$ **do**
3:     $\mathbf{g}_t^i = \mathbf{g}_t^i$                                            $\triangledown$ client side
4:     $\Delta_t^i = \mathcal{C}_\delta(\mathbf{g}_t^i - \mathbf{h}^i)$
5:     $\mathbf{h}_{t+1}^i = \mathbf{h}_t^i + \Delta_t^i$
6:     send to server: $\Delta_t^i$
7:     $\mathbf{x}_{t+1} := \mathbf{x}_t - \gamma \mathbf{h}_t$                       $\triangledown$ server side
8:     $\mathbf{h}_{t+1} = \mathbf{h}_t + \frac{1}{n} \sum_{i=1}^n \Delta_t^i$

---

Consider $F_t = f(\mathbf{x}_t) - f^\star$ and $H_t = \frac{1}{n} \sum_{i=1}^n \left\| \nabla f_i(x_t) - \mathbf{h}_t^i \right\|^2$. The following lemmas are simple modifications of the lemmas in (Li et al., 2021) and (Richtárik et al., 2021) in the presence of stochasticity. Therefore, we state them without proofs.

**Lemma 23** (Lemma 2 from Li et al. (2021)). *Let $f$ be $\mu$-strongly convex and $L$-smooth. Then iterates of* EF21-SGD *satisfy*

$$F_{t+1} \le (1 - \gamma\mu)F_t - \left( \frac{1}{2\gamma} - \frac{L}{2} \right) \mathbb{E}\left[ \|\mathbf{x}_{t+1} - \mathbf{x}_t\|^2 \right] + (2 - \delta)\gamma\sigma^2 + (1 - \delta)\gamma H_t. \tag{35}$$

**Lemma 24** (Lemma 7 from Richtárik et al. (2021)). *Let $f$ be $\mu$-strongly convex and $f_i$ be $L$-smooth for all $i \in [n]$. Then iterates of* EF21-SGD *satisfy*

$$\mathbb{E}_t\left[ H_{t+1} \right] \le \left( 1 - \frac{\delta}{4} \right) H_t + \frac{2L^2}{\delta} \mathbb{E}\left[ \|\mathbf{x}_{t+1} - \mathbf{x}_t\|^2 \right] + \frac{3\sigma^2}{\delta}. \tag{36}$$

Now consider $\Psi_t := F_t + a H_t$, where $a := \frac{100\gamma}{\delta}$. Putting these two lemmas together, we have:

**Lemma 25.** *Let $f$ be $\mu$-strongly convex and $f_i$ be $L$-smooth for all $i \in [n]$. Let $\gamma \le \frac{\delta}{100L}$. Then iterates of* EF21-SGD *satisfy*

$$\mathbb{E}_t\left[ \Psi_{t+1} \right] \le (1 - c)\Psi_t - \frac{40L}{\delta} \mathbb{E}_t\left[ \|\mathbf{x}_{t+1} - \mathbf{x}_t\|^2 \right] + \gamma\left( 2 + \frac{300}{\delta^2} \right)\sigma^2 \tag{37}$$

*where $c := \gamma\mu$.*

Solving the recursion, one can show that

$$\mu \Psi_{T+1} \le \frac{e^{-\gamma\mu(T+1)}}{\gamma} \Psi_0 + \left( 2 + \frac{300}{\delta^2} \right)\sigma^2$$

In particular, this means that $\Psi$ decreases to $\mathcal{O}(\frac{\sigma^2}{\delta^2 \mu})$ in $\mathcal{O}\left( \frac{L}{\delta\mu} \log \frac{F_0 \delta L}{\sigma^2 \mu} \right)$ rounds. Therefore, EF21-SGD can be used as warm up algorithm to find good approximation of $\mathbf{h}_\star^i$. As we can see, the output of EF21-SGD satisfies the restriction for Algorithm 3. In particular, we show that at any iteration of EF21-SGD, we have $\frac{1}{n} \sum_{i=1}^n \mathbb{E}\left[ \left\| \mathbf{h}_t^i - \nabla f_i(\mathbf{x}^\star) \right\| \right] \le 4L\Psi_t$ if $\gamma$ and $a$ are chosen as in Lemma 25.

**Lemma 26.** *If $f_i$ is $L$-smooth and convex, and $\mathbf{h}_t$ and $\mathbf{x}_t$ are generated by EF21 with $\gamma = \frac{\delta}{100L}$, then*

$$\frac{1}{n} \sum_{i=1}^n \mathbb{E}\left[ \left\| \mathbf{h}_t^i - \nabla f_i(\mathbf{x}^\star) \right\| \right] \le 4L\Psi_t$$

*where $\Psi_t = F_t + a H_t$ and $a = \frac{1}{\gamma}$.*

*Proof.*

$$\frac{1}{n}\sum_{i=1}^{n}\mathbb{E}\left[\left\|\mathbf{h}_t^i - \nabla f_i(\mathbf{x}^\star)\right\|^2\right] \leq \frac{2}{n}\sum_{i=1}^{n}\mathbb{E}\left[\left\|\mathbf{h}_t^i - \nabla f_i(\mathbf{x}_t)\right\|^2\right] + \frac{2}{n}\sum_{i=1}^{n}\mathbb{E}\left[\left\|\nabla f_i(\mathbf{x}_t) - \nabla f_i(\mathbf{x}^\star)\right\|^2\right]$$

$$\leq \frac{2}{n}\sum_{i=1}^{n}\mathbb{E}\left[\left\|\mathbf{h}_t^i - \nabla f_i(\mathbf{x}_t)\right\|^2\right]$$

$$+ \frac{4L}{n}\sum_{i=1}^{n}\left(f_i(\mathbf{x}_t) - f_i(\mathbf{x}^\star) - \langle\nabla f_i(\mathbf{x}^\star), \mathbf{x}_t - \mathbf{x}^\star\rangle\right)$$

$$= \frac{2}{n}\sum_{i=1}^{n}\mathbb{E}\left[\left\|\mathbf{h}_t^i - \nabla f_i(\mathbf{x}_t)\right\|^2\right] + 4L(f(\mathbf{x}_t) - f^\star)$$

$$= 2H_t + 4LF_t$$

$$\leq 4L\Psi_t \qquad \qquad \square$$

