# OpenReview forum: "EControl: Fast Distributed Optimization with Compression and Error Control"
_ICLR.cc/2024/Conference — ICLR 2024 poster_

### Official Review · Reviewer_6rfE · 2023-10-31

**Soundness:** 3 good
**Presentation:** 3 good
**Contribution:** 3 good
**Rating:** 6
**Confidence:** 3

**Summary:**

The paper proposes EControl algorithm that provably converges for strongly convex, convex, and nonconvex, with general contractive compression and with classic assumptions on the problem. It also gives experimental evaluations of EControl to show its efficacy
in practice.

**Strengths:**

New error compensation algorithm: EControl with theoritical gaurantees for strongly convex, general convex, and nonconvex functions
Empirical experiments to support the theoretical gaurantees and assess the efficacy of EControl.

**Weaknesses:**

see the section below (Questions)

**Questions:**

-In step 6 of EC-Ideal you don't have the extra parameter \eta that you have in EControl, can you explain why for EC-Ideal this is not needed. Will EC-Ideal have the same convergence guarantees if you add \eta in step 6, with the same values as for EControl?
-Bounded variance: This assumption contradicts strong convexity. I understand it is in several works in the literature. Take for instance F(x,y,z) = zx^2 + (1-z)y^2, z follows Bernoulli (1/2), then E_z(norm(\nabla F(x,y,z) - \nabla E_z(F(x,y,z)))^2) = (x-y)^2 and this is not bounded for all x and y in R.
-You mention in page 7 that "The asymptotic complexity of EControl in this regime with stochastic gradient is tight and cannot be improved,"  I understand for the depence on \epsilon, however can you tell why the dependence on other parameters can not be improved.
-In the experiments, how you relate \zeta to the noniddness, why for instance zeta equal to zero is iid and zeta >0 is non idd.
-Did you try \eta= delta/400 suggested by the theory in the experiments? why you need to fine tune it, since it is fixed by theory. Is only fixed \eta = \delta/400 that works in theory or many other values can work, this is not clear in the paper.
-Did you test with other compressors?

---

> ### Author Response · Authors · 2023-11-19
> **Response to Reviewer 6rfE (part I)**
>
> - ***In step 6 of EC-Ideal you don't have the extra parameter $\eta$ that you have in EControl, can you explain why for EC-Ideal this is not needed. Will EC-Ideal have the same convergence guarantees if you add $\eta$ in step 6, with the same values as for EControl?***
>
> In the EControl update, we notice that the error term $e$ is also injected into the $h$ update, and therefore the gradient estimator $h$ also carries some error information. We think this is the key reason why for EControl we need to introduce the new parameter $\eta$ to control the strength of the feedback signal because $g-h$ is already carrying some level of the feedback signal. In contrast, in the EC-ideal update, $h_\star$ is static and does not carry any error information. Therefore, in EC-Ideal, we compensate the full error information directly in each update. We also checked the modification of Lemma 15 with $\eta$ plugged in the update of $e_t^i$. Out derivations suggest to use constant $\eta$ which is close to $1$.
>
> - ***Bounded variance: This assumption contradicts strong convexity. I understand it is in several works in the literature. Take for instance $F(x,y,z) = zx^2 + (1-z)y^2, z$ follows Bernoulli (1/2), then $\mathbb{E}_z[||\nabla F(x,y,z) - \mathbb{E}_z \nabla F(x,y,z)||^2] = (x-y)^2$ and this is not bounded for all $x$ and $y$ in $\mathbb{R}$.***
>
> This is an interesting example. The bounded variance assumption in our work is the most used in the optimization literature to model the noise properties in the training. The classic example is when Gaussian noise with variance $\sigma^2$ is added to true gradients $\nabla f_i(x).$ We agree that it does not cover all possible noise models (e.g., the example the reviewer provided), but it is natural to assume when one analyses the properties of a new method. We leave more general noise models for future works.  (One such extension could be the noise model $||g^i(x) - \nabla f_i(x)||^2 \le M||\nabla f_i(x)||^2 + \sigma^2$. This noise model covers the example that the reviewer provided). See [9] and discussion therein for different noise model examples that one can use to extend the analysis of EControl to more complex noise models.
>
> - ***You mention in page 7 that "The asymptotic complexity of EControl in this regime with stochastic gradient is tight and cannot be improved," I understand for the dependence on $\epsilon$, however can you tell why the dependence on other parameters can not be improved.***
>
> With stochastic gradient, the asymptotically dominant term is $\frac{\sigma^2}{\mu n\varepsilon}$ in strongly quasi-convex regime and $\frac{\sigma^2}{n\varepsilon}$ in general convex and nonconvex regimes. These results match the lower bound provided in [7], up to a log term in $R_0$. We will update the manuscript to state it more precisely. Moreover, our results also match the lower bounds in the noiseless regime, i.e. $\sigma \to 0$, if we set $L_i=L$ for all $i\in[n]$,  where the last term in the rates becomes dominant.
>
> - ***In the experiments, how you relate $\zeta$ to the noniddness, why for instance $\zeta$ equal to zero is iid and $\zeta > 0$ is non idd.***
>
> There are several heterogeneity measures in the literature. For example, [8] introduces a measure $\overline{\zeta^2} = \frac{1}{n}\sum_{i=1}^n||\nabla f_i(x^*)||^2.$ In our experiments constant $\zeta^2$ is proportional $\overline{\zeta}^2$, therefore changing $\zeta^2$ we control the heterogeneity of the problem. In particular, it's easy to check that when $\zeta=0$ (and $\textbf{b}=0$, we will update the manuscript to clarify this), we have that $x^\star=0$ and $\nabla f_i(x^\star)=0,\forall i\in[n]$. The intuition behind this definition is the following: if local functions $f_i$ are close to each other, then the global minimizer $x^*$ is a good approximation of local minimizer ${\rm arg}\min f_i(x).$ Thus, $\overline{\zeta}^2$ is small and the problem is close to homogeneous setting and easier to solve. If local functions are very different from each other, then their local minimizers as well. Therefore, $\overline{\zeta}^2$ is large, i.e. the problem becomes harder to solve.
>
> - ***Did you try $\eta= \delta/400$ suggested by the theory in the experiments? why you need to fine tune it, since it is fixed by theory. Is only fixed $\eta = \delta/400 $ that works in theory or many other values can work, this is not clear in the paper.***
>
> The value $\eta\approx \frac{\delta}{400}$ is only a rough estimation in the sense that we did not optimize the inequalities in the technical lemmas to obtain the best possible absolute constant in theory. We present the lemmas this way so that they are easier to follow and understand.
>
> The important message here is that $\eta$ should scale with $\delta$, and the theory says any constant lower than $\frac{1}{400}$ works (and again, this upper bound can be further improved with more careful calculations), but doesn't specify what is the best constant in practice.

---

> > ### Comment · Reviewer_6rfE · 2023-11-20
> >
> > -Bounded variance assumption: I know that this hypothesis is in several works of literature. This is not enough to say that the hypothesis is too restrictive or not. For stochastic optimization or large finite sum, except special cases, you cannot have access to the exact gradient, all you can do is sample as in my example so no guarantee on the variance. Even if we can calculate the exact gradient, I don't see the need to add noise to this to have an SG, just use the gradient directly. So can you give some practical examples where we can have guarantees on this hypothesis.
> > -Heterogeneity measure: I don't see why $\bar{\zeta}$ is a good measure of it, especially in an overparameterized framework like in NN in general. If $f(x) = \sum f_i(x_i)$ where $x = (x_1,...)$ whatever f_is you can choose, whatever their dissimilarity $\bar{\zeta} = 0 $!
> > I checked [8], the comment on the heterogeneity measure $\bar{\zeta}$ in this work is as follows "\bar{\zeta} measures the diversity of functions fi. If all functions are the same , fi=fj , for all i,j, then \bar{\zeta}=0..$ are you really citing this as justification! Similarity between f_is does not mean equality!
> > -For you what is the similarity or dissimilarity between the f_is?
> > Please I don't want general diplomatic answers, I want mathematically well-founded arguments if you have. I don't want citations of papers to argue before you actually check what's in there.

---

> ### Author Response · Authors · 2023-11-19
> **Response to Reviewer 6rfE (part II)**
>
> [6] Yutong He, Xinmeng Huang, Yiming Chen, Wotao Yin, and Kun Yuan. Lower bounds and accelerated algorithms in distributed stochastic optimization with communication compression. arXiv preprint arXiv: 2305.07612, 2023
>
> [7] Sebastian U. Stich, Sai Praneeth Karimireddy, The Error-Feedback Framework: Better Rates for SGD with Delayed Gradients and Compressed Communication, JMLR, 2020.
>
> [8] Anastasia Koloskova, Nicolas Loizou, Sadra Boreiri, Martin Jaggi, Sebastian U. Stich, A Unified Theory of Decentralized SGD with Changing Topology and Local Updates, ICML 2020, 2020.
>
> [9] Ahmed Khaled and Peter Richtárik. Better theory for SGD in the nonconvex world. arXiv preprint arXiv:2002.03329, 2020.

---

> ### Author Response · Authors · 2023-11-22
> **Response to Reviewer 6rfE**
>
> - ***Can you give some practical examples where we can have guarantees on this hypothesis?***
>
> Please note that the bounded variance assumption is a classic model [13] used in the analysis of stochastic optimization methods. We list below some examples of ML problems where this condition holds.
>
> -- Any function with a bounded gradient satisfies this assumption. One of the instances is the logistic regression function $$\frac{1}{n}\sum_{i=1}^n\log\left(1+\exp(-b_i \cdot \textbf{a}_i^\top\textbf{x})\right).$$
>
> -- [14] show that this assumption is also satisfied in Reinforcement Learning applications for bounded policies, see Proposition 1 in [14].
>
> We agree with the reviewer that models with growth conditions (e.g., $\|g^i(x) - \nabla f_i(x)\|^2 \le \sigma^2 + M\|\nabla f(x)\|^2$) cover a broader class of functions. It would be interesting to extend the analysis to these settings.
>
> The example of adding Gaussian noise has not been properly described. We do not claim that one should add Gaussian noise to gradients. Please excuse any confusion this might have caused.
>
> [13] A. Nemirosky, D. Yudin, Problem complexity and method efficiency in optimization, J. Wiley @ Sons, New York, 1983
>
> [14] Ilyas Fatkhullin, Anas Barakat, Anastasia Kireeva, Niao He, Stochastic Policy Gradient Methods: Improved Sample Complexity for Fisher-non-degenerate Policies, ICML 2023.
>
> - ***For you what is the similarity or dissimilarity between the $f_i$'s?***
>
> Please note, that our algorithm (EControl) does not rely on any heterogeneity assumption. However, the convergence of EC is impacted by the gradient dissimilarity (in theory [see Table 1 in our manuscript], and this dependency is also verified in our experiments).
>
> Because, the earlier work (e.g. [6, 8]) used the quantity $\bar\zeta^2 = \frac{1}{n}\sum_{i=1}^n||\nabla f_i(x^\star)||^2$, we measure (and control) this quantity for our experiments on the synthetic least squares problem.
>
> If we address all your questions, please kindly consider maintaining your original rating.

---

> ### Author Response · Authors · 2023-11-22
> **Response to Reviewer 6rfE**
>
> We’ve updated the description of the experiment to clarify that, with the synthetic least square problem, we are numerically demonstrating that EControl’s convergence is indeed independent of the gradient dissimilarity (which is a limiting factor for many of convergence analysis of the prior works on communication compression, see also section 2.2 for a more detailed discussion), and hence corroborate our theoretical findings. We would like to thank the reviewer for pointing out this ambiguity in our wordings.

---

### Official Review · Reviewer_iMbi · 2023-11-01

**Soundness:** 3 good
**Presentation:** 2 fair
**Contribution:** 3 good
**Rating:** 6
**Confidence:** 3

**Summary:**

This paper presents EControl, a novel algorithm introduced to address communication compression challenges in modern distributed training. EControl utilizes contractive compressors to reduce communication overhead, effectively mitigating issues like unstable convergence and compression bias commonly associated with naive implementations. Notably, EControl's standout feature lies in its ability to regulate error compensation by controlling the feedback signal's strength, making it a valuable solution for data heterogeneous scenarios.

Unlike previous methods, EControl dispenses with impractical assumptions such as bounded gradients or data homogeneity. It demonstrates rapid convergence rates in standard convex and nonconvex settings, showcasing its adaptability and robustness. Through comprehensive numerical evaluations, the paper substantiates the effectiveness of EControl, highlighting its potential to revolutionize distributed training by alleviating communication overhead challenges and facilitating more efficient and stable model training processes.

**Strengths:**

The proposed method EControl does not need the assumptions such as large batchsizes and bounded gradient, it also enjoys linearly speedup, therefore, it overcomes the shortcomings of previous error-compensation framework.

**Weaknesses:**

While the importance of the crucial parameter $\eta$ in EControl is discussed in section 4, its significance could benefit from a more comprehensive explanation within the main body of the paper. I would strongly recommend the inclusion of lemmas and inequalities to elucidate the pivotal role of $\eta$. Additionally, providing similars lemmas into how $h_t^i$ approximates gradient information and how the strength of the feedback signal is controlled would enhance the clarity and understanding of the proposed method.

**Questions:**

Same questions related to the issues on specific lemmas or inequlities asked in "Weaknesses".

---

> ### Author Response · Authors · 2023-11-19
> **Response to Reviewer iMbi**
>
> - ***While the importance of the crucial parameter $\eta$  in EControl is discussed in section 4, its significance could benefit from a more comprehensive explanation within the main body of the paper. I would strongly recommend the inclusion of lemmas and inequalities to elucidate the pivotal role of $\eta$.***
>
> The parameter $\eta$ plays a crucial role in stabilizing the convergence. First, we practically observe that setting $\eta=1$ leads to divergence even for toy example; see section C in the appendix for more details. This means that it has to be set smaller than $1$. Next, as the reviewer NamU mentioned,  with $\eta=0$ and special initialization EControl covers EF21 algorithm which does not converge in the stochastic setting. Therefore, we need to set $\eta$ small, but still positive. Intuitively, $\eta$ controls the contribution of error term in the training. If it is too large, then the gradient approximation becomes too far from the true gradient. If it is too small, then we do not keep enough information from previous iterations. From more technical point of view, $\eta$ controls how the error $E_t=\frac{1}{n}\sum_{i=1}^n||e_t^i||^2$ contributes in the descent lemmas. It turns out that the choice $\eta \sim \delta$ allows to decrease all terms in the Lyapunov function proportionally to each other.
>
> In the main body of the manuscript, we aim to provide a more intuitive and high level description of the method and the idea of error control. The specific statements of the lemma might be too technical for this purpose. We will add a more comprehensive discussion of the parameter $\eta$ and its role in the lemmas in the appendix, and in the main body referring interested readers to these discussions.
>
> - ***Additionally, providing similar lemmas into how $h_t^i$ approximates gradient information and how the strength of the feedback signal is controlled would enhance the clarity and understanding of the proposed method.***
>
> EControl is designed so that the norm difference between $h_t^i$ and stochastic gradients $g_t^i$ is controled. Indeed, we now show that it is controlled by our Lyapunov function:
>         $$\frac{1}{n}\sum_{i=1}^n\mathbb{E}[||g_t^i-h_t^i||^2] \leq 2\frac{1}{n}\sum_{i=1}^n\mathbb{E}[||\eta e_t^i+g_t^i-h_t^i||^2]+2\frac{1}{n}\sum_{i=1}^n\eta^2\mathbb{E}[||e_t^i||^2].$$
>         We note that $\frac{1}{n}\sum_{i=1}^n\mathbb{E}[||\eta e_t^i+g_t^i-h_t^i||^2]=H_t$ and $\frac{1}{n}\sum_{i=1}^n\eta^2\mathbb{E}[||e_t^i||^2]=\eta^2E_t$. With the choice $\eta \sim \delta$, $H_t+\eta^2E_t$ is proportional to the $aH_t + bE_t$ term in the Lyapunov function we use in the analysis. Taking into account the convergence guarantees for the Lyapunov function, the above derivations show that we do control the norm difference between $h_t^i$ and $g_t^i$ not allowing it to blow up.

---

> > ### Comment · Reviewer_iMbi · 2023-11-22
> >
> > Thank you for the response. I don't have any other comments.

---

### Official Review · Reviewer_NamU · 2023-11-01

**Soundness:** 3 good
**Presentation:** 3 good
**Contribution:** 2 fair
**Rating:** 6
**Confidence:** 3

**Summary:**

This paper proposes EControl, a novel mechanism that can regulate error compensation by controlling the strength of the feedback signal. Theoretical analysis is provided for EControl in strongly convex, general convex, and nonconvex settings without any additional assumptions on the problem or data heterogeneity. The experiments show that the proposed algorithm out-performs the baselines in large-scale vision tasks.

**Strengths:**

1. This paper proposes EControl, a novel mechanism that can regulate error compensation by controlling the strength of the feedback signal.

2. Theoretical analysis is provided for EControl in strongly convex, general convex, and nonconvex settings without any additional assumptions on the problem or data heterogeneity.

3. The experiments show that the proposed algorithm out-performs the baselines in large-scale vision tasks.

**Weaknesses:**

1. It is unknown how to combine EControl with momentum SGD. The original EF has well-developed momentum variants. EF21 has the variant of EF21-SGDM. Although the experiments include EF21-SGDM, there is no EControl + momentum provided.

2. There seems to be a strong connection between EControl and EF21. If we the compressed version of $h$ in the initialization $h_0^i = \mathcal{C}(g_0^i)$, and take $\eta = 0$, then EControl will be equivalent to EF21. In other words, with a little bit twist in the initialization, EF21 can be viewed as a special case of EControl. However, I haven't found any discussion about the relationship between EControl and EF21 like this. Furthermore, I recommend to add some ablation experiments where $\eta$ varies and gets tuned gradually closer to 0, which I guess will give a training loss curve closer to EF21.

**Questions:**

1. Is there a variant of EControl which is combined with momentum?

2. Could the authors give a more detailed discussion about the (theoretical) connection and comparison between EControl and EF21?

---

> ### Author Response · Authors · 2023-11-19
> **Response to Reviewer NamU**
>
> - ***Is there a variant of EControl that is combined with momentum?***
>
> The goal of the project was to create a method which solves the main challenges of Error Compensation: making it provably works in all standard setting, the convergence improves with $n$ and relies on standard smoothness and/or convexity assumption. Designing a momentum variant of EControl is certainly an interesting direction to consider, and in the literature there exists a number of different ways to incorporate momentum into distributed training algorithms, for example, local momentum e.g. [1] [2] [3]; server side momemtum e.g. [4]; and even quasi-global momentum [5]. We will investigate which one would be most suitable for EControl in practice and theory.
>
> - ***Could the authors give a more detailed discussion about the (theoretical) connection and comparison between EControl and EF21?***
>
> Setting $\eta=0$ and special initialization EControl recovers EF21. Therefore, it is important to keep $\eta$ positive since EF21 does not work well in the stochastic case: it does not converge if the batch size is small; the convergence does not improve with $n.$ In contrast, EControl does have all these properties. Moreover, it provably works in all standard regimes.
>
> From a more technical point of view, in the EF21 paper, they are controlling the distance between $h_t^i$ and the full local gradient $\nabla f_i(x_t)$ (see also our Appendix G).  In contrast, EControl is designed so that the norm difference between $h_t^i$ and stochastic gradients $g_t^i$ is controled. Indeed, we now show that it is controlled by our Lyapunov function:
>         $$\frac{1}{n}\sum_{i=1}^n\mathbb{E}[||g_t^i-h_t^i||^2] \leq 2\frac{1}{n}\sum_{i=1}^n\mathbb{E}[||\eta e_t^i+g_t^i-h_t^i||^2]+2\frac{1}{n}\sum_{i=1}^n\eta^2\mathbb{E}[||e_t^i||^2].$$
>
> We note that $\frac{1}{n}\sum_{i=1}^n\mathbb{E}[||\eta e_t^i+g_t^i-h_t^i||^2]=H_t$ and $\frac{1}{n}\sum_{i=1}^n\eta^2\mathbb{E}[|| e_t^i||^2]=\eta^2E_t$, where $H_t$ and $E_t$ are defined in Appendix A. With the choice $\eta \sim \delta$, $H_t+\eta^2E_t$ is proportional to the $aH_t + bE_t$ term in the Lyapunov function we use in the analysis. Taking into account the convergence guarantees for the Lyapunov function, the above derivations show that we do control the norm difference between $h_t^i$ and $g_t^i$ not allowing it to blow up.
>
> We will discuss these differences and connections between EControl and EF21 in the appendix, and in the main body referring interested readers to these discussions. Moreover, we have conducted an experiment on decreasing $\eta$ to zero, and showed that in the stochastic setting, the performance of EControl degrades and gets closer to EF21 as $\eta$ goes to zero. We will add it to the appendix as well.
>
> [1] Ilyas Fatkhullin, Alexander Tyurin, Peter Richtárik, Momentum Provably Improves Error Feedback!, arXiv preprint arXiv: 2305.15155.
>
> [2] Assran, M., Loizou, N., Ballas, N., and Rabbat, M. Stochastic gradient push for distributed deep learning. In International Conference on Machine Learning, pp. 344–353. PMLR, 2019.
>
> [3] Koloskova, A., Lin, T., Stich, S. U., and Jaggi, M. Decentralized deep learning with arbitrary communication compression. In International Conference on Learning Representations, 2020a.
>
> [4] Ilyas Fatkhullin, Igor Sokolov, Eduard Gorbunov, Zhize Li, and Peter Richtárik. EF21 with bells \& whistles: Practical algorithmic extensions of modern error feedback. arXiv preprint arXiv:2110.03294, 2021.
>
> [5] Tao Lin, Sai Praneeth Karimireddy, Sebastian U Stich, and Martin Jaggi. Quasi-global momentum: accelerating decentralized deep learning on heterogeneous data. International Conference on Machine Learning, pages 6654–6665, 2021.

---

### Official Review · Reviewer_8MAp · 2023-11-03

**Soundness:** 3 good
**Presentation:** 3 good
**Contribution:** 3 good
**Rating:** 8
**Confidence:** 3

**Summary:**

This work proposes a new EC algorithm for distributed training with communication compression. Unlike traditional error compensation methods where we only add the previous compression back to the compressed gradient, in EControl authors achieve a faster convergence rate by controlling the strength of the feedback signal. Authors proved the superiority of EControl from both theoretical and experimental side.

**Strengths:**

The algorithm design looks smart to me. By starting from the EC-ideal, the convergence rate of the compressed SGD could achieve optimal asymptotic complexity. Afterwards authors further extend this idea to general cases where h^* cannot be attained.

**Weaknesses:**

No

**Questions:**

No

---

### Author Response · Authors · 2023-11-19
**General Response to all Reviewers**

Dear reviewers,

Thanks a lot for your thorough reviews of the paper and valuable comments. We have made the following updates to the paper following your suggestions:

- we add a discussion on how the distance between $g_t^i$ and $h_t^i$ are controlled in Remark 15 at the end of Appendix B. This is referred to in the main body;

- we add a technical discussion on the role of $\eta$ in the lemmas in Appendix C.1, and a discussion on the connection between EControl and EF21 and how as $\eta\to0$ the performance degrades to that of EF21 in the stochastic case in Appendix C.3. These are referred to in the main body as well;

- we updated the discussion on the complexity of EControl in section 6 to be more precise;

- we fixed the description of the synthetic least squares experiments.

---

### Meta-Review · Area_Chair_UaSL · 2023-12-13

**Metareview:**

The paper introduces EControl, an algorithm that converges for strongly convex, convex, and nonconvex functions with general contractive compression. EControl uses contractive compressors to reduce communication overhead and mitigate issues like unstable convergence and compression bias. EControl's standout feature is its ability to regulate error compensation by controlling the feedback signal's strength, making it a valuable solution for data heterogeneous scenarios.

Strengths: The proposed method EControl overcomes the shortcomings of previous error-compensation frameworks by not requiring assumptions such as large batch sizes and bounded gradients. EControl also enjoys linearly speedup. This method is a new error compensation algorithm with theoretical guarantees for strongly convex, general convex, and nonconvex functions. Empirical experiments support the theoretical guarantees and assess the efficacy of EControl.

Weaknesses: The current EControl does not support moments, which is a limitation that Reviewer NamU has pointed out.

**Justification For Why Not Higher Score:**

See the weakness above.

**Justification For Why Not Lower Score:**

The theoretical guarantees offered by the proposed method are superior to those of previous work.

---

### Decision · Program_Chairs · 2024-01-16

Accept (poster)